# Transforming sustainable plant proteins into high performance lubricating microgels

Ben Kew[1], Melvin Holmes ●[1] ✉, Evangelos Liamas[1,2], Rammile Ettelaie[1], Simon D. Connell ●[3] ✉, Daniele Dini ●[4] & Anwesha Sarkar ●[1] ✉

With the resource-intensive meat industry accounting for over 50% of food-linked emissions, plant protein consumption is an inevitable need of the hour. Despite its significance, the key barrier to adoption of plant proteins is their astringent off-sensation, typically associated with high friction and consequently poor lubrication performance. Herein, we demonstrate that by transforming plant proteins into physically cross-linked microgels, it is possible to improve their lubricity remarkably, dependent on their volume fractions, as evidenced by combining tribology using biomimetic tongue-like surface with atomic force microscopy, dynamic light scattering, rheology and adsorption measurements. Experimental findings which are fully supported by numerical modelling reveal that these non-lipidic microgels not only decrease boundary friction by an order of magnitude as compared to native protein but also replicate the lubrication performance of a 20:80 oil/water emulsion. These plant protein microgels offer a much-needed platform to design the next-generation of healthy, palatable and sustainable foods.

There is arguably no bigger existential challenge for consideration of food scientists than ensuring the security of safe, affordable, palatable, healthy and environmentally-sustainable nutrients to feed the growing human population, achieving the United Nations' Sustainable Development Goals (SDGs) to help alleviate the perils of hunger while respecting the planet's environmental boundaries. Global greenhouse gas (GHG) emissions from the production of food have been estimated to be >18 Gt $CO_2$eq yr$^{-1}$ (one-third of the total human-caused GHG emissions[1]), of which those produced by animal-based foods (57%) have been calculated to be nearly twice as those of plant-based foods (29%)[2]. To ensure the continued supply of safe, pleasurable and healthy food, while reducing net GHG emissions, the transition from animal to plant-based foods is a much-needed endeavour. As a result, food manufacturers have reacted by incorporating and increasing plant proteins in their food products. However, the adoption of plant proteins at a population scale is restricted and require vast time-scales in product development due to their sensorial-functionality issues[3–5]. It is now well-evidenced that one of the primary barriers to adoption of plant protein is their negative astringent i.e., dry, puckering, non-juicy perception. Oral lubrication serves as a well-acknowledged in vitro proxy for quantifying friction-related mouthfeel characteristics[6]. Multiscale tribology measurements across laboratories supported by sensory trials have revealed that plant proteins increase oral friction due to particle-like protein-protein aggregation and jamming as well as interaction with saliva, in contrast to dairy proteins[7–11]. Such lubrication failure or delubrication[9] highlights a major issue if plant proteins are to be used instead of conventional animal proteins. The adverse textural modifications and development of these astringent, drying characteristics would reduce consumer acceptability of plant protein-rich foods. For instance, it is the lubrication behaviour governed by coalescence of fat, which generates desirable low-friction creamy mouthfeel[12] whilst in non-lipidic, fat mimetics, hydration lubrication and molecular ball-bearing

[1]Food Colloids and Processing Group, School of Food Science and Nutrition, University of Leeds, Leeds LS2 9JT, UK. [2]Unilever Research & Development Port Sunlight, Quarry Road East, Bebington, Merseyside CH63 3JW, UK. [3]Molecular and Nanoscale Physics Group, School of Physics and Astronomy, University of Leeds, Leeds LS2 9JT, UK. [4]Department of Mechanical Engineering, Imperial College London, London SW7 2AZ, UK. ✉e-mail: prcmjh@leeds.ac.uk; s.d.a.connell@leeds.ac.uk; A.Sarkar@leeds.ac.uk

mechanisms[13] are often cited to be the key mechanisms to achieve optimal lubrication performance[8].

Proteins themselves have also been attempted as a fat mimetic for producing greater satiety, lower calorie foods. With less than half the calories of fat per gram (4 kcal vs 9 kcal) offering highest satiety per calorie[14], as well as viscosity enhancing properties, proteins are ideal macronutrients to substitute fat; however, the lack of understanding and maximising in lubrication performance has become the major limitation. Of recent considerable interest for fat mimetics and the improved texture of proteins are the creation of protein microgel structures. Protein microgels, are crosslinked, swollen, percolating protein networks which are sheared down to discrete micron- or nanometric-sized soft colloidal particles. These microgels mostly consist of water (85–95%) by weight with moduli ranging from 0.1–10 kPa[15]. Animal protein-based, polysaccharide as well as synthetic microgels such as whey protein-based, polyacrylic acid-based and carrageenan-based microgels have demonstrated varying degree of lubricity[16–18] depending upon volume fraction and elasticity of particles entrained in the contact. Often such lubrication mechanisms have been governed by viscosity modification with microgels acting as physical surface separators[17,18] or by so-called (often debated) ball bearing mechanisms[13,19]. Nevertheless, protein-based fat mimetic research has been largely restricted to dairy proteins, with rare attention being given to plant proteins[5]. With the intense pressure to replace animal proteins with more sustainable, ethical, hypoallergenic proteins, plant proteins are occupying the research landscape rapidly and methods to improve performance in food are needed to avoid unacceptability and associated food wastage. Additionally, plant proteins also face intra-variability within the protein type, where both natural climate conditions and industrial processing can result in non-standardised proteins with a range of poor solubility, limited functionality and poor hydrating ability[20,21] resulting in oral dryness. These present major challenges for plant proteins to be used as a food ingredient where microgelation can be a much-needed structuring platform to standardise and overcome these oral frictional hurdles, which is the key question this study answers. Therefore, lubrication evidence on microgelation of plant proteins are imperative before such microgels can be applied as a highly functional, pleasurable ingredient for designing next generation plant protein-reformulated foods[22].

Herein, we use a combination of experimental and theoretical approaches to demonstrate that engineering physically cross-linked, soft, sub-micron sized plant-based microgels offers higher lubrication performance as compared to their parent native protein counterparts. Remarkably, these fabricated microgels allow an order of magnitude reduction in boundary frictional forces as compared to the proteins. Using a complementary suite of techniques such as rheology, tribology, quartz crystal microbalance with dissipation monitoring (QCM-D), dynamic light scattering (DLS), atomic force microscopy (AFM) and mathematical modelling, we show that such microgelation of plant proteins renders lubricity similar to that of an oil/water (O/W) emulsion without using any additional lipidic substance. To aid a more accurate in vivo prediction of oral surface induced frictional behaviour, a biomimetic 3D tongue-like surface with similar deformability, topography and surface wettability to that of real human tongue was utilised for tribological measurements. These designed plant-based microgels provide a platform of food ingredients to enhance palatability and functionality, thus, the improved design of the next generation protein-based, healthy, tasty and sustainable diets in order to accelerate the transition to plant-based foods.

## Results

### Structure of microgels across scales
We start by assessing the size, morphology and stability of the plant protein microgels. This study utilised pea protein and potato protein as typical exemplar plant proteins, which take the form of globular multimer storage proteins. Pea proteins are composed mostly of 11 S and 7 S globulins[23] and frequently reported for limited aqueous solubility[8,21,23] resulting in high friction and sensorial astringency[9,24]. On the other hand, potato proteins are composed of globular, glycoprotein i.e., patatins[25], with comparatively higher fractions of soluble protein[8,25,26] but still suffer from astringency issues[8,24] due to high surface hydrophobicity[27]. A blend of the proteins were also investigated to determine any synergistic or detrimental effect on lubricity. The latter approach also provides a way to enhance the amino acid profile of possible formulations through protein complementation, a crucial consideration in non-animal protein diets[28]. Four types of microgels were fabricated utilising a top-down methodology involving thermal gelation-induced physical crosslinking of these otherwise sedimenting (Supplementary Fig. 1) plant proteins resulting in a percolating, viscoelastic protein-based hydrogels followed by controlled shearing into microgels (Fig. 1, see detailed preparation in method section). In order to attain different elasticities of these microgels, which are known to be important in lubrication[18], and the high protein concentrations typically found in tribology studies involving high levels of friction[8], the concentration of the solutions was adjusted to promote gel formation with varying properties which offer an opportunity to study the microgelation process and its effectiveness for lubrication in plant proteins (see details in method section).

Microgels fabricated using pea protein concentrate (15.0 wt% total protein, PPM15), potato protein isolate at two concentrations (5.0 and 10.0 wt% total protein PoPM5, PoPM10) and a mixed plant protein (pea protein concentrate 7.5 wt% total protein, potato protein isolate 5.0 wt% total protein, PPM7.5:PoPM5) at volume fractions ($\Phi$) 10–70 vol% are shown in Fig. 2a. We find, as expected, an increase in opacity with higher $\Phi$ (Fig. 2a), especially true for microgels able to scatter more light thanks to their larger hydrodynamic diameter ($d_H$), as obtained using dynamic light scattering (DLS) (Fig. 2b). The brown pigmentation from potato aromatics[29] and orange hue from pea is not surprising and results from the presence of natural carotenoids[30]. At the nanoscale, microgels can be described as soft, gel-bead like particles 100–1000 nm in size with an ability to swell or deswell in altered solvent conditions. These gelled particles are likely to consist of a hydrated protein core with a hydrated water-reservoir like shell as a result of a concentration gradient of protein-water from the core to the outer surface of the microgel, displaying diffuse brush topography with protrusions from the core[15] (Fig. 1).

Individual microgels (PPM15, PoPM5, PoPM10) display $d_H$ ranging from 67 to 204 nm possessing low polydispersity index (PDI) in contrast with the mixed microgels (Fig. 2b), PPM7.5:PoPM5 which show two distinct peaks where the peak <100 nm likely represents the potato protein system. This is likely due to the different denaturation temperatures with potato protein gelling at 60 °C[26] and pea protein at > 85 °C[31]. Thus, these two proteins do not complex into one microgel remaining as separate microgel entities perhaps with minor sub-unit complexation as $d_H$ remain similar to those individual microgels. Such $d_H$ values are typical to other reported microgels from both animal and plant protein-based sources[18,32–34] with their differences in size being dependant on protein conformation, concentration, solubility and water holding capacity besides processing variables. When comparing to native proteins i.e., non-microgelled plant protein isolates/concentrates, these often produce more variation in size as they are highly polydisperse, aggregated and may need filtering due to sedimentation[8,35]. For instance, microgels even after a month of storage show no sedimentation in comparison to the non-microgelled counterparts that sediment within few hours (Supplementary Fig. 1) where there is little change in particle size (Supplementary Table 1). Even when the microgels were further processed such as those simulating food processing (e.g., thermally treated at

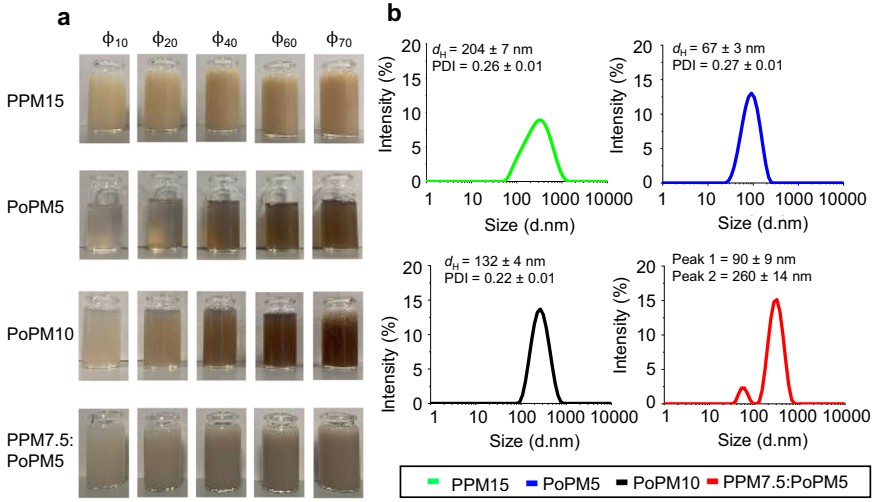

**Fig. 1 | Schematic illustration of microgelation of native plant proteins.** Visual representation of microgelation procedure. Native plant proteins are highly aggregated causing functional and sensory problems in food design. By hydrating them with water and thermally gelling using hydrophobic interactions, hydrogen bonding and disulphide-based covalent crosslinking occurring without any added crosslinking agents, the native plant proteins act as connecting particulate points in a highly percolating hydrogel network, which is then converted into gel-like particles via controlled homogenisation consisting of 5–15 wt% protein and 85–95 wt% water. These microgels remove functional issues associated with native protein allowing for improved functional application of plant proteins in food.

**Fig. 2 | Particle size of plant protein microgels.** Visual images (**a**) showing various degrees of opacity and size distribution obtained using dynamic light scattering (DLS) (**b**) using pea protein concentrate to form a 15.0 wt% total protein microgel, (PPM15), potato protein isolate to form a 5.0 wt% total protein microgel (PoPM5), potato protein isolate to form a 10.0 wt% total protein microgel, (PoPM10), and using a mixture of pea protein concentrate at 7.5 wt% total protein and potato protein isolate at 5.0 wt% total protein microgel (PPM7.5:PoPM5) at volume fractions ($\phi$) 10–70 vol%, at 25.0 °C. Insets in (**b**) shows the mean hydrodynamic diameter ($d_H$) and polydispersity index (PDI) values. Results are plotted as average of six measurements on triplicate samples ($n = 6 \times 3$).

90 °C for 30 min), no marked change was observed in hydrodynamic diameter or polydispersity index (Supplementary Table 2) highlighting the excellent thermodynamic stability of these microgels.

To investigate microgel morphology in more detail, atomic force microscopy (AFM) was employed to image fully hydrated microgels, shown in Fig. 3 with corresponding particle size distributions. All microgels are in the same general size range of 50–200 nm as found by DLS. (Fig. 2b).

The mixed PPM:PoPM (Fig. 3d) system displays two distinctly sized population of microgels which could be the result of pea and potato protein forming microgels individually as discussed previously in DLS. PoPM5 (Fig. 3b) is remarkably similar in size in both techniques, 67 nm (DLS) vs 73 nm (AFM). In the other samples the AFM size was slightly smaller than DLS; for PoPM10 (Fig. 3c) the diameters DLS:AFM were 132 nm:109 nm, and for the mixed PPM:PoPM were 90 nm: 78 nm

for the first peak, and 260 nm: 158 nm for the second. This reduction in size can be explained by noting that the loosely structured brush-like features extends from the surface of microgels and occupies a larger hydrodynamic volume. We speculate the increased overestimation from DLS could be related to the hydration shell whilst AFM measures the protein core, this explains the similar size of PoPM5 and larger size of PoPM10 as more protein have an extended influence on the hydrodynamic diameter, latter otained using DLS. PPM15 (Fig. 3a) showed a larger discrepancy, 204 nm (DLS) vs 79 nm (AFM). This could be explained by the presence of a small number of polydispersed large particles or aggregates in this sample, which could skew the DLS measurement to a higher value (these aggregates were not seen in the distributions of PoPM5 and PoPM10) or show the presence of a highly hydrated shell spanning far from the core. Shape analysis (Supplementary Fig. 2) showed the majority of particles were spherical or near

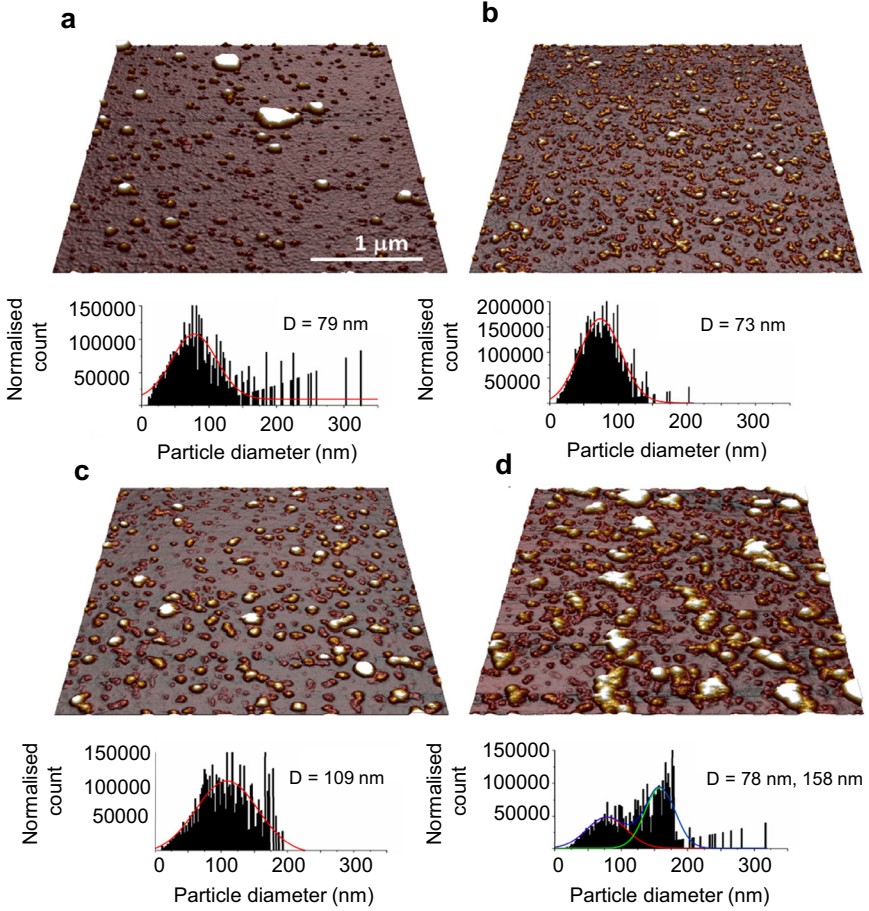

**Fig. 3 | Images of plant protein microgels on silicon under buffer.** Topographic images and respective histograms showing diameters of aqueous dispersions of protein microgels prepared using (**a**) pea protein concentrate to form a 15.0 wt% total protein microgel, (PPM15), (**b**) potato protein isolate to form a 5.0 wt% total protein microgel (PoPM5), (**c**) potato protein isolate to form a 10.0 wt% total protein microgel, (PoPM10), and (**d**) a mixture of pea protein concentrate at 7.5 wt% total protein and potato protein isolate at 5.0 wt% total protein to form microgel (PPM7.5:PoPM5).

spherical with an aspect ratio <2:1, although with increasing aspect ratio with size, which is probably explained by the larger particles being aggregates. Overall microgels take on a smooth and spherical shape showing convexed spreading on the surface (Fig. 3a–d), which has also been observed for synthetic and whey protein microgels in previous studies[18,36].

## Viscosity of microgels

Microgels are cited in literature as viscosity modifiers with a range of thinning/ thickening behaviour[37] with the ability to influence lubrication performance particularly in the fluid film regimes by altering viscosity as perceived at the macroscale[18]. Before characterising viscosity, it is nevertheless important to understand the stiffness of the microgels which may influence their viscous dissipation. For example, higher storage modulus (G′) of the parent protein gels may translate into higher viscosities of the microgel dispersion derived from these gels and associated fluid film lubrictaion[18]. We assume microgels are sub-micrometric units of the parent hydrogels and thus possess the same elasticity. To quantify this, oscillatory shear rheology (see temperature ramp and frequency sweeps in Supplementary Figures 3 and 4, respectively) was performed on the parent protein hydrogels prior to shearing to obtain G′ and loss (G″) modulus (Fig. 4a) and large scale deformation tests were performed (Supplementary Fig. 5) to calculate the Young's modulus (Fig. 4b). Typically, the higher the protein content of the microgel, the higher the G′. For instance, G′ of the parent gels for PoPI5 (~800 Pa), was an order of magnitude lower than that of PoPI10 (~8500 Pa), explaining the value of PoPM10 viscosity observed

in Fig. 4c compared to the softer, easily compressible microgels contributing to lower viscosity, which is true of PoPM5.

Most plant protein-based microgels in this study display Newtonian behaviour (Fig. 4c) as viscosity is independent of shear rates suggesting a non-interactive network of microgels with the exception at Φ = 60–70% where there is a clear shear thinning behaviour. Particularly, in the case of PoPM10 Φ = 60–70% when compared to native protein of the same protein concentration, in which a strong pseudoplastic behaviour is observed, where particle interactions dominate and sharp increases of viscosity with increased Φ is evident (Supplementary Fig. 6c). In general, such rheological behaviour of microgels is in sharp contrast to those of the native proteins where especially PoPI5.5, PoPI7.0 and PPC5.2.5:PoPI3.5 show shear thinning behaviour and an initial aggregated particle network (Supplementary Fig. 6b–d). This leads us to propose an interesting conjecture, microgelation in plant proteins offer a desirable structural route to make the system much less aggregated, create a better control of size as compared to the parent protein, irrespective of their type.

The change in solution viscosity as a function of volume fraction (Φ = 10–70%) is shown at low shear rates (Fig. 4c) and at orally relevant shear rates (Fig. 4d). Although as expected the magnitude differs, the microgel viscosity shows similar dependency on volume fraction, irrespective of shear rates (Fig. 4c, d). We find in line with our expectation that as Φ increases, an increase in viscosity is observed for all microgels with PoPM10 displaying more pronounced increases at a low Φ = 40% compared to all other microgels, with PoPM5 showing little viscosity changes throughout.

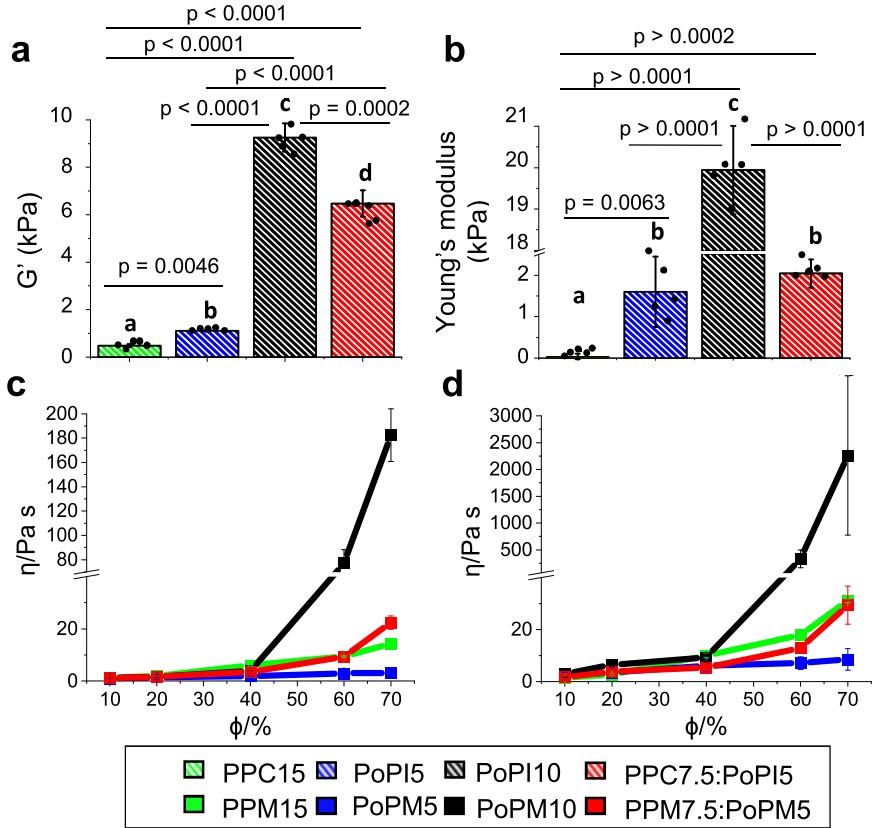

**Fig. 4 | Rheological properties of the parent plant protein gels and volume fraction-dependent apparent viscosities of the microgels.** Storage modulus (**a**) and Young's modulus (**b**) of parent plant protein gels with apparent viscosities (η) of microgels prepared using pea protein concentrate to form a 15.0 wt% total protein microgel, (PPM15), potato protein isolate to form a 5.0 wt% total protein microgel (PoPM5), potato protein isolate to form a 10.0 wt% total protein microgel, (PoPM10), and using a mixture of pea protein concentrate at 7.5 wt% total protein and potato protein isolate at 5.0 wt% total protein microgel (PPM7.5:PoPM5) with corresponding storage modulus (G′) of parent plant protein gels (**a**) as a function of volume fractions (ϕ) at pH 7.0 at shear rates of (**c**) 0.1 s−1 and (**d**) 50 s−1, the latter representing orally relevant shear rates performed at 37 °C. Data was recorded with

shear rate increasing from 0.1 to 50 s−1, Figures a–b display means and standard deviations of 5 measurements on triplicate samples ($n = 5 \times 3$) where statistical analysis was performed using two tailed unpaired Student's t-test with differing lower-case letters in the same bar chart indicate a statistically significant difference ($p < 0.05$). Figures (**c**, **d**) shows the mean of 6 measurements on triplicate samples ($n = 6 \times 3$) with error bars representing standard deviations. The original temperature ramp and frequency sweeps of the parent heat set gelled proteins are shown in Supplementary Figs. 3 and 4, respectively. The true stress-strain curves are shown in Supplementary Fig. 5 from which the Young's moduli are computed. Original flow curves for the microgel dispersions at each volume fractions are shown in Supplementary Fig. 6.

An interesting anomaly is observed with PPM15, which despite containing the highest protein content, has a low viscosity. This is explained by the modulus of PPC15 (Fig. 4a, b) which provides further evidence of the pea protein's inability to gel effectively due to low solubility (~30%[8]) in aqueous media and form microgels of high volume fractions. Additionally, viscosity flow curves of the microgels at each volume fraction (Supplementary Fig. 6) reveals further discrepancies where PPM15 has less than half of G′ of PoPM5 (Fig. 4a), yet viscosity is higher in Φ = 10–20 vol% and similar for other Φ. When analysing mixtures of these proteins (PPC7.5:PoPI5) both proteins are able to synergistically contribute to the structure as moduli increases vastly compared to PoPM5 and PPM15 which are both ranked as the second strongest hydrogels (Fig. 4a). Strikingly, PPM15 displays close resemblance to the viscous behaviour of PPM7.5:PoPM5 (see Fig. 4c, d and Supplementary Fig. 6) despite having a markedly lower modulus (Fig. 4a, b) ($p < 0.05$). This shows the gel structure is held primarily by the gelling of PoPM5 (Fig. 4b) but when microgelled, the parent gels breaks down into two separate particles with viscosity influenced by pea protein moieties as PoPM5 has little viscosity influence (Fig. 4c, d). Overall the evidence suggests that when homogenised separation of the two protein microgels from the gelled matrix occurs, which is also evident from

the size distribution (Figs. 2b and 3d), PPM7.5:PoPM5 result in two disparately sized populations of microgels.

For model hard spheres, a random packing limit of Φ = 64% causes a steep increase in viscosity as mobility of particles in solution is restricted. It is well known that microgels do not follow this model, instead they show such changes often at higher than Φ = 64% due to their ability to swell, interpenetrate, deform and their soft nature[38]. PoPM10 packing limit is met earlier, most likely due to the high content of soluble protein (100%- solubility at pH 7[8]) and higher modulus (see Fig. 4a, b) allowing for the markedly higher viscosities recorded.

In summary, the microgels convert the native proteins into much less aggregated structures and the viscosity show Φ dependency only above Φ = 40% except for PoPM5, where the viscosity is independent of Φ.

## Tribological performance

In the field of food soft-tribology, the friction coefficient has important connotations in respect to sensory mouthfeel. More specifically, low friction coefficients are found to correlate to pleasurable smoothness, creaminess and fat-like properties, where conversely higher values correspond to roughness, astringency and off-mouthfeel[6]. Friction coefficient is used here as a measure of lubrication performance for all

the created plant protein microgels ($\Phi = 10–70\%$). These measurements were compared for native protein counterparts (equivalent protein to $\Phi = 70\%$) and a 20:80 O/W emulsion to observe both improvements in lubrication versus native proteins and similarities of microgel with O/W lubrication, determined between steel and soft PDMS tribocontact surfaces (Fig. 4, see Supplementary Table 3 for statistical comparison). The viscosity component was considered in the lubrication performance through scaling, here obtained by multiplying the friction coefficient ($\mu$) on the abscissa by high shear plateau viscosity at $1000\,s^{-1}$. For reference, original $\mu$ against entrainment speeds across $10^{-5}–10^2$ is provided in Supplementary Fig. 7. The buffer, native proteins and emulsions display a typical Stribeck curve, with boundary regime depicted at $0.001\,Pa\,m$, mixed regime at $0.01–0.1\,Pa\,m$ and elastohydrodynamic regime at and above $1.0\,Pa\,m$. The microgels (Fig. 5) showed immediate onset of mixed regime even at very low speed x viscosity values of $0.04\,Pa\,m$, irrespective of the protein type or volume fractions, highlighting their ultra-lubricating behaviour.

Of most importance is that in comparison to native proteins all microgels had significantly lower friction when compared at $0.1\,Pa\,m$ ($p < 0.01$) throughout the mixed regime. Remarkably PPM15 obtained more than an order of magnitude decrease in friction in comparison to native protein ($\mu = 0.006–0.01$ compared to $\mu = 0.14$, ($p < 0.01$)) with other microgels obtaining at least a 5 fold reduction (refer to Supplementary Table 3). Additionally, microgels display significantly lower friction from $0.1\,Pa\,m$ through to $0.3\,Pa\,m$ and microgels at $\Phi = 10–40\%$ resembled or outperformed the lubrication performance of the O/W emulsion ($\mu = 0.03$ at $0.1\,Pa\,m$) ($p < 0.05$).

When comparing differences in volume fraction (Figs. 5a–d, 1–3), microgels were highly efficient in decreasing friction at all $\Phi$. The exception was PoPM10, where at $\Phi = 70\%$, significantly higher friction was exhibited than all other microgels measured at $0.1\,Pa\,m$ ($p < 0.05$). This was likely due to a very high viscosity of PoPM10, which resulted in potential aggregation or jamming and consequently inability to effectively entrain between the surfaces in contact. A small but significant ($p < 0.05$) increase in friction was also observed when increasing the volume fraction from $\Phi = 10\%$ to $\Phi = 70\%$ for PPM15 and PPM7.5:PoPM5, which may be attributed to particle-particle adhesion. This would mean the formation of larger aggregates[39] as previously seen by ref. [7]. The lowest friction amongst all samples was obtained for PPM15 at $\Phi = 10\%$ ($\mu = 0.0057$ at $0.1\,Pa\,m$) with friction coefficient half of that of other microgels at the same $\Phi$. At $\Phi = 40\%$ $\mu$ values were between $0.006–0.01$ whilst at $\Phi = 70\%$ all microgels achieve friction of $0.01$ with the exception of PoPM10 ($\mu = 0.035$) where despite such varied protein concentrations ($3.5–10.5\,wt\%$) microgelation was able to standardise friction amongst all proteins.

Overall at $0.1\,Pa\,m$, the $\mu$ values irrespective of volume fractions are one-to-two orders of magnitude lower than buffer ($\mu = 0.74$) with microgels closely resembling the friction coefficients of O/W emulsions (with exception of PoPM $\Phi = 70\%$). This highlights the overall effectiveness that microgelation has on protein to provide lubrication properties even resembling that of O/W emulsions without any lipidic additive, thus eliminating high friction issues associated with the plant proteins. To our knowledge, such extreme improvement in lubrication performance of delubricating[9] plant proteins (down to lowest $\mu$ values of $<0.005$ in many cases) (Fig. 5a-d1, a-c2 at $0.3\,Pa\,m$), achieved by transforming them into microgels has never been reported in literature to date.

### Theoretical fit of lubrication performance of microgels

The Stribeck curve is often adopted to model the total friction coefficient ($\mu_{tot}$) with increasing entrainment speed $U$ or, the Hersey number, a nondimensional variable formed by the product of the dynamic viscosity $\eta$ of the fluid with speed $U$ divided by the normal load $F_N$ per length of the contact[28]. The Stribeck curve takes the form shown in Eqs. (1–3)[28–30].

$$\mu_{tot} = \mu_{EHL} + \left( \frac{\mu_b - \mu_{EHL}}{1 + (U\eta/B)^m} \right) \quad (1)$$

$$\mu_{EHL} = k(U\eta)^n \quad (2)$$

$$\mu_b = h(U\eta)^l \quad (3)$$

where ($k$, $n$) and ($h$, $l$) are the elastohydrodynamic regime (EHL) and boundary layer power law coefficients and index, respectively, $B$ relates to the threshold value of $U\eta$ for boundary friction and $m$ the mixed regime exponent. These parameters must be empirically determined. To establish an alternative model, we proceed by calculating expressions for the torque experienced through friction of a rotational disc of radius $R$ (m) on the surface of a fluid with load $F_N$ (N) and, equivalently, for the torque which relates to the fluid properties under rotational shear i.e., angular speed $\omega$, viscosity $\eta$ (N s m$^{-2}$) and height $h$ (m) of the fluid between the contact surfaces. The coefficient of friction $\mu$ is defined as the ratio of the resistive frictional force $F$ opposing the motion of two surfaces to a normal compressive force $F_N$, namely $\mu = F/F_N$. Thus,

$$F = \sigma A$$

where, $\sigma$ denotes the stress and $A$ is the contact area, and,

$$\sigma = \frac{F}{A} = \frac{\mu F_N}{A} = \frac{\mu F_N}{2\pi R^2}$$

Integrating to obtain the total torque $T$ over the whole surface of disks i.e., with increasing radial distance $r$, we have,

$$T = \int_0^R 2\pi r^2 \sigma dr = \int_0^R \frac{2\pi \mu F_N r^2}{2\pi R^2} dr = \frac{\mu F_N}{R^2} \left[ \frac{r^3}{3} \right]_0^R = \frac{\mu F_N R}{3} \quad (4)$$

Similarly, using the stress-rate of strain relation we may write,

$$\sigma = \eta \left( \frac{dv}{dz} \right) = \eta \frac{\omega r}{h}$$

Let us consider the Torque $T$.

$$dT = dFr$$

$$dF = \sigma dA$$

$$dA = 2\pi r dr$$

$$dT = 2\pi r^2 \sigma dr = \frac{2\pi r^3 \eta \omega dr}{h}$$

$$T = \int_0^R \frac{2\pi r^3 \eta \omega}{h} dr = \frac{2\pi \eta \omega}{h} \left[ \frac{r^4}{4} \right]_0^R = \frac{\pi \eta \omega R^4}{2h} \quad (5)$$

Equating Eqs. (4) and (5)

$$\mu = \frac{3\pi \eta \omega R^3}{2F_N h} \quad (6)$$

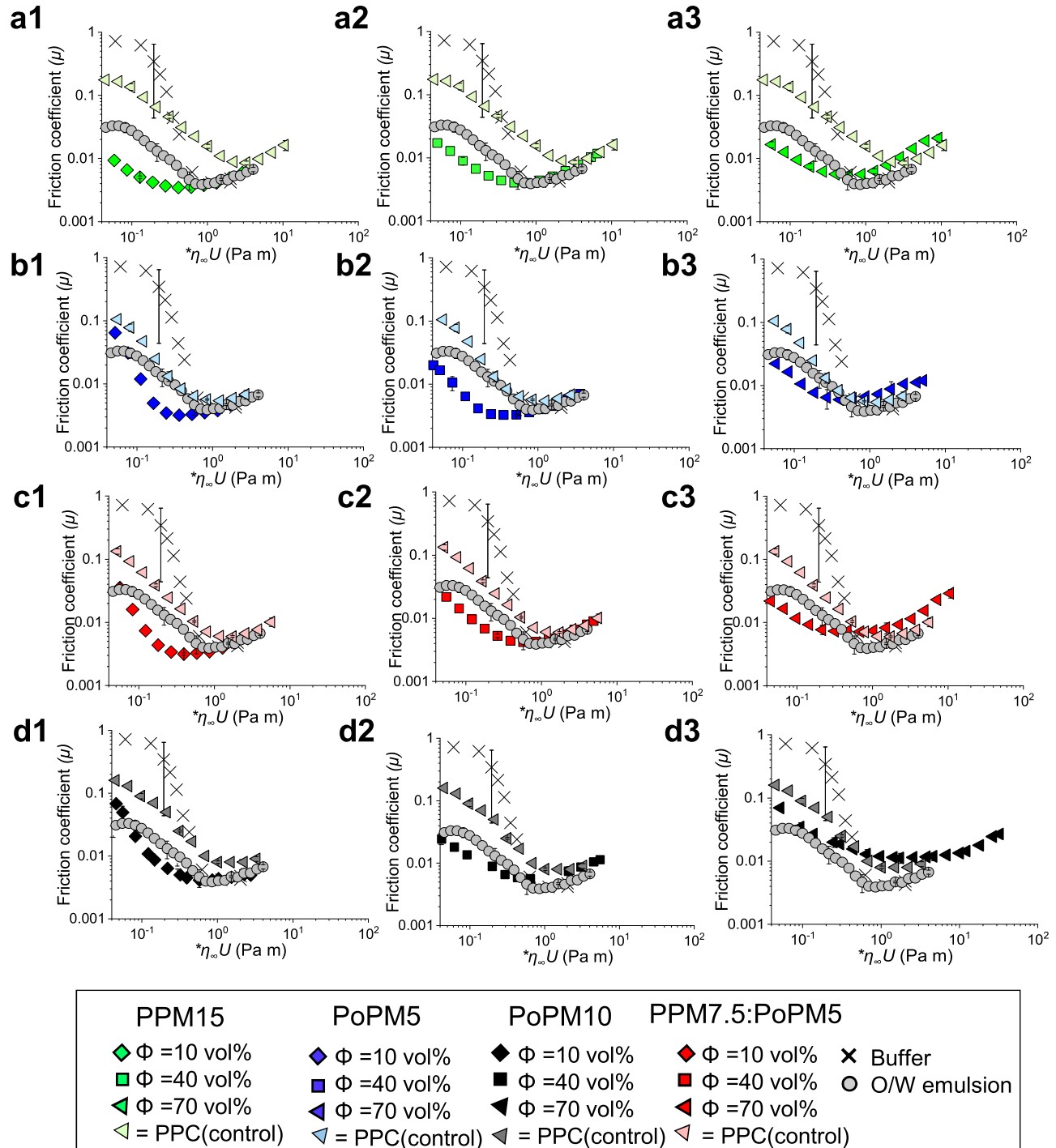

**Fig. 5 | Stribeck curves in hard-soft contact surfaces in presence of plant protein microgels.** Tribological performance of steel ball on PDMS surfaces in the presence of plant protein microgels, native plant protein (matched protein content for Φ = 70 vol% with numbers displayed relating to total protein content) or oil-in-water emulsion. Friction coefficient (μ) as a function of entrainment speed (U) scaled with high rate viscosity (η ∞ = 1000 s − 1) in the presence of plant protein microgels prepared using (**a1**–**a3**) pea protein concentrate to form a 15.0 wt% total protein microgel, (PPM15), (**b1**–**b3**) potato protein isolate to form a 5.0 wt% total protein microgel (PoPM5), (**c1**–**c3**)), potato protein isolate to form a 10.0 wt% total protein microgel, (PoPM10), and (**d1**–**d3**) using a mixture of pea protein concentrate at 7.5 wt% total protein and potato protein isolate at 5.0 wt% total protein microgel (PPM7.5:PoPM5) with 1, 2 and 3 showing increased volume fractions from 10 to 70 vol%, respectively. Frictional responses of the plant proteins at the highest concentration and 20 wt% oil-in-water emulsion (O/W emulsion) and buffer are included in each graph (a-d) as controls. Results are plotted as average of six repeat measurements on triplicate samples ($n = 6 \times 3$) with error bars representing standard deviations. Statistical comparison of mean at 0.1 Pa m is shown in Supplementary Table 3. Original friction coefficient versus entrainment speed curves for the microgel dispersions at each volume fractions are shown in Supplementary Fig. 7.

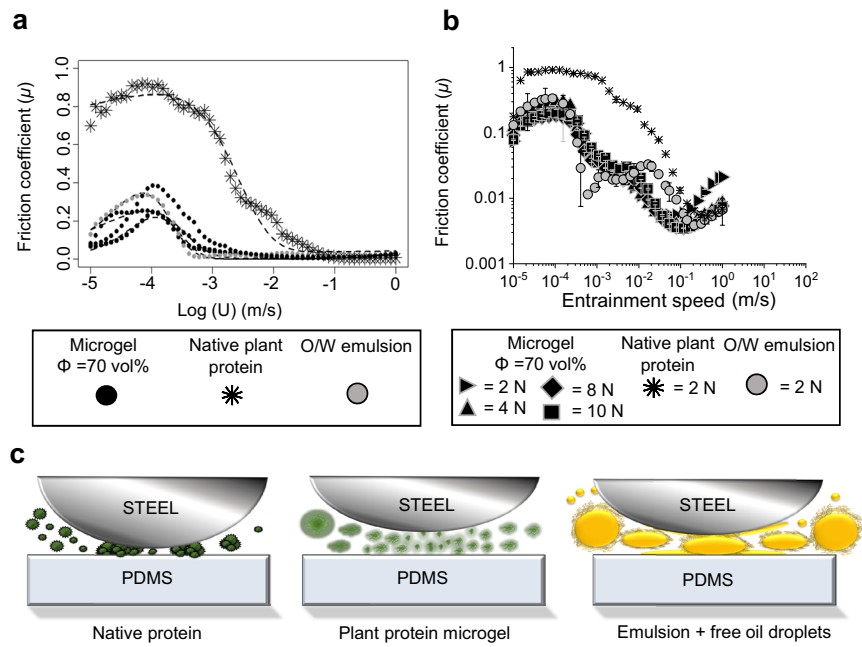

**Fig. 6 | Mechanism of lubrication performance of plant protein microgels in hard-soft contact surfaces.** Tribological performance of steel ball on PDMS contact surfaces showing (**a**) theoretical modelling of lubrication performance at a load of 2 N of exemplar plant protein microgels (pea, potato and mixed pea and potato microgel) showing close resemblance to the emulsions as opposed to the large friction coefficients obtained in presence of the native protein. Here the dashed lines show the best theoretical fit using Eq. 11 and (**b**) load dependency of microgels as compared to the native protein (matched protein content for Φ = 70 vol%) with 20:80 O/W emulsion as control with (**c**) schematic illustration of microgel performance as compared to native protein in hard-soft contacts. Friction coefficient (μ) is plotted as a function of entrainment speed (U). Results are plotted as average of three repeat measurements on triplicate measurements ($n = 3 \times 3$) with error bars representing standard deviations.

Assuming the fluid follows an Ostwald−de Waele law as can be expected for microgel dispersions being a power-law fluid,

$$\sigma = K\left(\frac{\partial v}{\partial z}\right)^n = K\left(\frac{\omega r}{h}\right)^n \tag{7}$$

$$dT = 2\pi r^2 \sigma dr = 2\pi K r^2 \left(\frac{\omega r}{h}\right)^n dr = 2\pi K r^{n+2}\left(\frac{\omega}{h}\right)^n dr$$

$$T = \int_0^R 2\pi K r^{n+2}\left(\frac{\omega}{h}\right)^n dr = 2\pi K \left(\frac{\omega}{h}\right)^n \left[\frac{r^{n+3}}{n+3}\right]_0^R = 2\pi K \left(\frac{\omega}{h}\right)^n \frac{R^{n+3}}{n+3}$$

$$\frac{\mu F_N R}{3} = 2\pi K \left(\frac{\omega}{h}\right)^n \frac{R^{n+3}}{n+3}$$

$$\mu = \frac{6\pi K}{F_N}\left(\frac{\omega}{h}\right)^n \frac{R^{n+2}}{n+3} = \frac{6\pi K R^2}{F_N(n+3)}\left(\frac{\omega R}{h}\right)^n \tag{8}$$

In lubrication theory, the specific height λ between the two surfaces can be related to the surface roughness and asperities $\sigma_1^2$ and $\sigma_2^2$ of the two surfaces, and the fixed surface separation $h$ as shown in Eq. (9). Here, we assume that the asperity $\sigma_1$ of one surface is fixed i.e., belongs to the tribometer device and that of the other is determined by the sample specific height, which will be dependent upon the entrainment speed $U$.

$$\lambda = \frac{h}{\sqrt{\sigma_1^2 + \sigma_2^2}} \tag{9}$$

In the current context, since we have conducted experiments under a constant applied load, we do not speculate on the nature of

the dependence of the applied load (or in other words normal force) i.e., $F_N$, and simply confine the effective height to be dependent upon entrainment speed. We suggest the specific height $\lambda$ of the fluid layer takes a functional dependence on the speed $U$, and which may be estimated empirically. A suitable functional form for this function must be such that it is asymptotically maximal/minimal at zero/high entrainment speeds, respectively, and therefore, must take a sigmoidal form similar to that of Eq. (9) or equivalently other forms which exhibit this behaviour such as a logistic equation or the adopted form used in this work, the Gompertz Eq. (10). Accordingly, by substitution of Eq. (10) into Eq. (8), we arrive at Eq. (11), where the parameters, $a$, $b$, $c$ and $K$ are empirically determined. Figure 6a shows the experimental data of the friction coefficient against Log $U$ and the associated empirically fitted model described by Eq. (11). Models were fitted using non-linear solvers which optimally minimised summed residuals.

$$h(U, F_N) = ae^{-e^{b-cU}} \tag{10}$$

Here $a$ is the maximum value of the asymptote, $b$ is the displacement along the velocity axis and $c$ the growth rate scaled by U. Thus,

$$\mu = \frac{6\pi K R^2}{F_N(n+3)}\left(\frac{U}{h(U)}\right)^n \tag{11}$$

In Fig. 6a, we show the theoretical modelling of lubrication performance using Eq. (11) at a load of 2.0 N of exemplar plant protein microgels (pea, potato and mixed pea and potato) showing close resemblance to the emulsion as opposed to the large friction coefficients obtained in presence of native protein. In Supplementary Fig. 8, we show the normalised (to the initial level) friction coefficient to scale and remove the dependence upon the applied load. It may be

observed that the native protein requires higher speeds before a reduction in normalised coefficient is observed in contrast to the microgel and emulsion sample. We suggest that friction-velocity profiles may be fitted using Eq. (11) and that in the boundary regime, the specific height $\lambda$ is negligible resulting in high friction and, this reduces under shear in the mixed regime as the effective fluid height increases as the fluid is entrained and, which will attain a minimal asymptotic level when the specific fluid height reaches its maximum and the system is in the elastohydrodynamic region. In the latter case, this will be the case when the available 'free' fluid within the microgel has been released from the structure increasing the depth of the fluid layer and reducing asperities. This demonstrates the effective height of microgels vs native proteins is crucial in lubrication, comparing microgel to native protein there is a clear reduction in friction coefficient owing to their water saturated structure demonstrating a striking improvement in lubrication (Fig. 6a). Furthermore, a similar frictional response is obtained to a O/W emulsion enhancing the extended use of microgels as a fat mimetic demonstrating clear improvements.

Next, we determine whether microgels release fluids upon tribological stress. A load-dependence procedure was followed where friction under increasing loads (2–10 N) were measured (Fig. 6b). It is now clear that microgels as a result of increased load show no significant differences in lubrication throughout boundary and mixed regimes (0.001–0.1). Microgels remained resilient and when compared to native structures, frictional curves did not converge to the native plant protein even at 10 N. This suggests even under high normal force microgels still provide support, despite undergoing changes, whereby under load they act as mini-reservoirs of water which promote hydration lubrication[40] and behaving differently to native protein.

In order to understand the microgel lubrication mechanism, the deformation was computed by calculating indentation from Hertz theory of contact points[41] similarly performed previously for microgels and emulsion[42]. This model allows us to understand how particles and microgels break, squeeze and flow to support lubrication of the load. The normal load supported by lubricant ($W_L$) and the individual particle/microgel ($W_p$) as well as the reduced elastic modulus formed between particle/microgel and PDMS surface ($E^*$) were determined. We also estimate the number of particles/microgels with radius R, forming a monolayer inside the contact with an effective fraction ($\phi_p$) covering contact area ($a_{TP}$), (refer to Supplementary Information 1 for theoretical analysis).

The relative indentation can be expressed as Eq. (12):

$$\frac{\delta}{R} = \left(\frac{a_H}{R}\right)^2 - \frac{4}{3\pi(1-v^2)}\left(\frac{a_H}{R}\right)f\left(\frac{a_H}{R}\right) \tag{12}$$

where, $v$ is the Poisson ratio estimated for gels and the ratio $a_H/R$ is the radius of contact which is independent of $R$, relating to fraction of surface covered by particles/microgels $\phi_p$, expressed as Eq. (13)

$$\frac{a_H}{R} = \left(\frac{3W_L}{4\phi_p E^* a_{TP}^2}\right)^{1/3} \tag{13}$$

We also estimated the entrainment force of the microgels being dragged into contact using Stokes drag Eq. (14):

$$F_d = 6\pi R\eta U \tag{14}$$

Based on theoretical and experimental results, we have generated indentation and drag force values (Table 1) with a subsequent schematic for lubricity (Fig. 5c). We observe that all microgel lubricants support a high percentage (>90%) of the load ($W_L$) resulting in the decrease in friction coefficient (Table 1). However, due to the difference in viscosity component as well as the elasticity between the

**Table 1 | Theoretical mechanism of lubrication. Calculation of relative Indentation and drag force of the emulsion droplets and microgels**

| Lubricant | $W_L$ (%) | $\delta R^*$ | $\eta$ at 1.0 s⁻¹ (Pa s) | $W_p$ (N) | $F_d$ (N) |
|---|---|---|---|---|---|
| PPM15 | 95.2 | 1.18 | 0.03 | $7.7 \times 10^{-5}$ | $2.0 \times 10^{-10}$ |
| PoPM5 | 92.2 | 1.57 | 0.01 | $6.1 \times 10^{-5}$ | $2.0 \times 10^{-11}$ |
| PoPM10 | 93.5 | 3.23 | 2.26 | 0.01 | $8.4 \times 10^{-9}$ |
| PPM7.5:PoPM5.0 | 95.0 | 2.85 | 0.03 | 0.026 | $2.5 \times 10^{-10}$ |
| O/W emulsion | 96.4 | 0.62 | 0.0001 | $4.5 \times 10^{-4}$ | $5.0 \times 10^{-11}$ |

emulsion and microgels, the drag force, as well as the load supported by individual particles/microgels, differed. From the indentation data, it is clear that emulsion droplets most likely deform into an elliptical shape under contact ($\delta R^* = 0.62$) in contrast to the microgels that are rather fully deformed ($\delta R^* > 1$), where $R^*$ is the reduced radius. This suggests that breakdown of the microgels would promote surface coverage between asperities, allow weeping of water into the contact, therefore increasing localised viscosity enhancing lubricity (Fig. 6C), as a parallel mechanism to the coalescence of oil observed in O/W emulsions to generate low friction by separating contact. In addition, the gel material may further swell generating hydration lubricity observed from adsorption measurements when exposed to free water resulting in a steric viscoelastic hydrated layer separating contact (Supplementary Fig. 9). Overall, the one-to-two orders of magnitude higher drag force (except for PoPM5) and $W_p$ for microgels as compared to the emulsions drive the microgels to the tribological contact and allow supporting the load by virtue of their elasticity and viscosity. This then explains the microgel lubricity mechanism.

## Lubrication using biomimetic tongue surface

Oral tribology has provided significant advances in friction mediated sensory responses, which is supported by over a decade of correlating tribology to real sensory attributes[6,43,44]. In oral tribology, the paradigm has historically been using smooth, hydrophobic, PDMS tribopairs of high elastic modulus (~2 MPa) as a surface to represent the human tongue. However, such materials differ in their wettability, contact pressure and topography from a real human tongue[45,46], thus hindering true friction-sensory correlation in complex soft materials. In the pursuit of improving accuracy and reliability of in vitro oral mouthfeel measurements, specific attention is given to the development of biorelevant surfaces[45]. Therefore to establish mechanistic understanding behind friction-mouthfeel associations, lubrication of plant microgels and O/W emulsion was measured between a bespoke biomimetic 3D tongue-like surface on steel contact (Fig. 7), created from a 3D printed mould which incorporates papillae size and spatial distribution, elasticity and wetting properties of human tongue[45]. For reference, exact values and statistical comparison can be found in Supplementary Table 4. Unlike PDMS-steel contact (Supplementary Fig. 7) looking at range of entrainment speed, the biomimetic tongue-steel contact focuses on boundary friction coefficients.

Similarly to PDMS-Steel contact, we measure excellent lubrication for microgels showing similarities to O/W emulsion where we observe a non-significant or significantly improved friction at all volume fractions of PPM15, $\Phi = 70\%$ of PoPM5 and PoPM10 and $\Phi = 10-40\%$ of PPM7.5:PPM5, again reinforcing such excellent lubricity of plant protein microgels. Also observed from Fig. 5c3, higher speeds result in increased friction in presence of PoPM10 (Fig. 7c3) arising from high solution viscosity, limiting easy entrainment between contacts.

This technique does present several differences arising from the properties of the tongue-surface specifically, its wettability, topography and deformability which may incur high frictional sensitivity in the boundary friction. In particular, PoPM5 $\Phi = 10$, PoPM10 $\Phi = 10$ and PPM7.5:PoPM5 $\Phi = 70$ were found to have frictions higher than O/W at

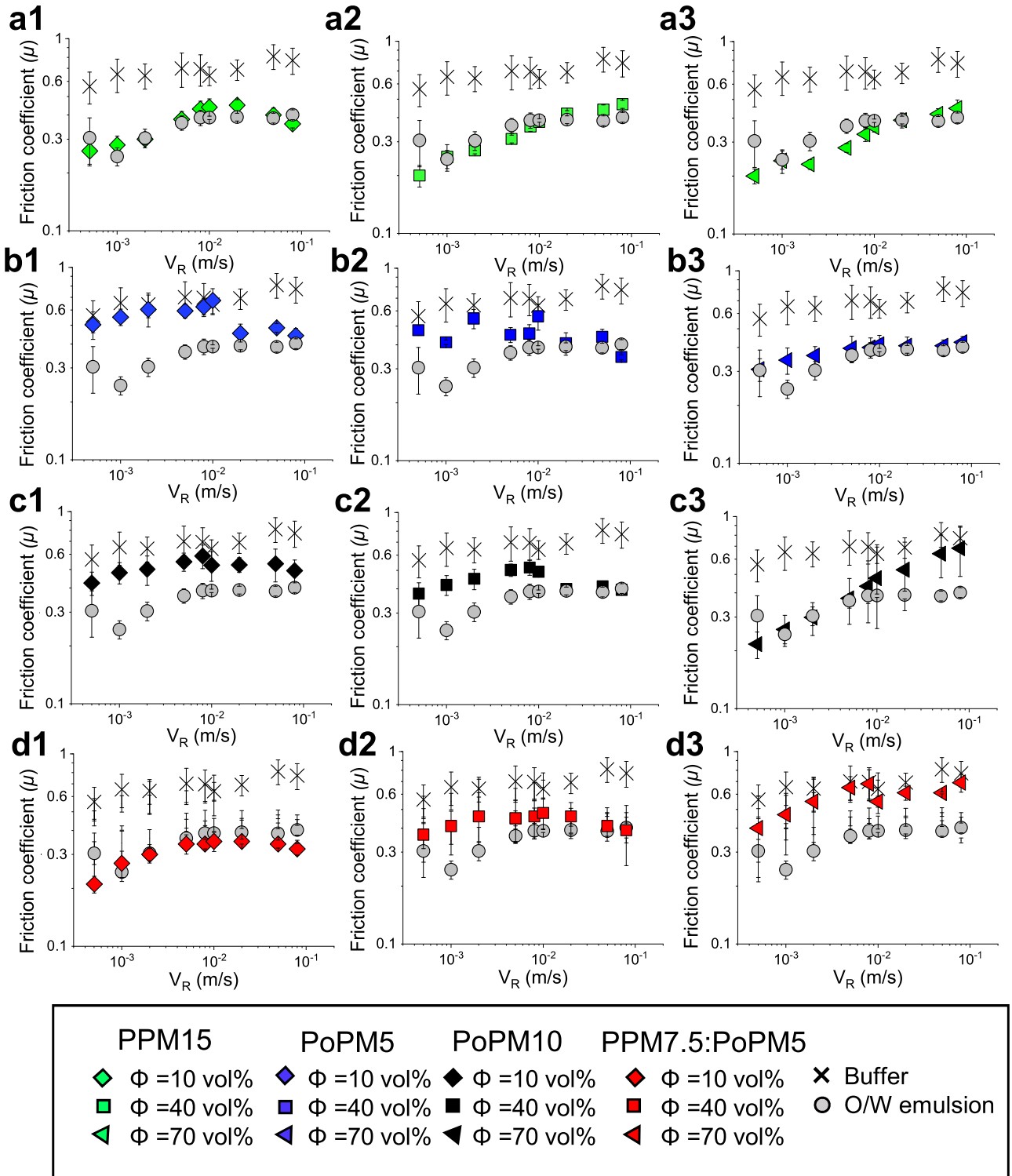

**Fig. 7 | Tribological performance in 3D biomimetic tongue-like surfaces.** Tribological performance of 3D-printed biomimetic tongue-like polymeric surfaces in presence of plant protein microgels or oil-in-water emulsions. Friction coefficient (μ) as a function of linear speed (VR) in the presence of plant protein microgels prepared using (**a1**–**a3**) pea protein concentrate to form a 15.0 wt% total protein microgel, (PPM15), (**b1**–**b3**) potato protein isolate to form a 5.0 wt% total protein microgel (PoPM5), (**c1**–**c3**)), potato protein isolate to form a 10.0 wt% total protein microgel, (PoPM10), and (**d1**–**d3**) using a mixture of pea protein concentrate at 7.5 wt% total protein and potato protein isolate at 5.0 wt% total protein microgel (PPM7.5:PoPM5), respectively. Frictional responses of 20 wt% oil-in-water emulsion (O/W emulsion) and buffer are included in each graphs (**a**–**d**) as controls. Results are plotted as average of six measurements ($n = 6 \times 3$) with error bars representing standard deviations. Statistical comparison of means at each lubrication regimes is shown in Supplementary Table 4.

**Table 2 | Quantitative assessment of binding of plant protein microgels to hydrophobic surface**

| Protein microgel | Hydrated mass (mg m$^{-2}$) | $-\Delta D/\Delta f$ |
|---|---|---|
| PPM15 | 11.1 +/− 2.6[a] | 0.08 +/− 0.050[a,b] |
| PoPM5 | 8.4 +/− 0.3[b] | 0.10 +/− 0.003[b] |
| PoPM10 | 24.0 +/− 2.1[c] | 0.12 +/− 0.070[b,c] |
| PPM7.5:PoPM5 | 16.7 +/− 6.6[a,b,c] | 0.06 +/− 0.020[a,c] |

Mean and standard deviation (SD) obtained from three repeat measurements on triplicate samples ($n = 3 \times 3$) of the hydrated mass and viscoelasticity ($-\Delta D/\Delta f$) plant protein microgels prepared using pea protein concentrate to form a 15.0 wt% total protein microgel, (PPM15), potato protein isolate to form a 5.0 wt% total protein microgel (PoPM5), potato protein isolate to form a 10.0 wt% total protein microgel, (PoPM10), and using a mixture of pea protein concentrate at 7.5 wt% total protein and potato protein isolate at 5.0 wt% total protein microgel (PPM7.5:PoPM5). Data were obtained using 3rd, 5th, 7th and 11th overtones. Different lower-case letters in the same column indicate a statistically significant difference ($p < 0.05$). Original frequency and dissipation shift data are shown in Supplementary Fig. 9.
[a–c]Parameters denoted with the same lower case subscripts do not differ statistically at the confidence of $p \geq 0.05$.

lower $V_R$ ($p > 0.05$). We speculate a higher $\delta R^*$ and lower $d_H$ may contribute to an initial ineffective lubrication for potato protein microgels as there may not be sufficient levels of microgel for successful entrainment. However, at higher $V_R$ the friction reduces where a non-significant difference in $\mu$ to those in O/W emulsion is observed ($p > 0.05$). For PPM7.5:PoPM5.0, an increase in volume fraction results in increased friction. As this dispersion is made up of two sets of microgels of different moduli and size, a build-up may occur at higher concentrations at the papillae promoting a jamming behaviour opposed to uniformly hydrated ball-bearing-type lubrication if microgel of one protein type[47] is present. This phenomenon is not observed in glass and PDMS as it is likely the flat surface contact allows separate microgels to flow over one another without disruption, improving lubricity.

To relate adsorption and lubrication properties, hydrated mass and adsorption kinetics of the protein microgels were analysed using QCM-D on PDMS coated sensors (see Supplementary Fig. 9 for frequency and dissipation shifts). The hydrated mass has been inversely correlated to mixed and hydrodynamic regimes depicting the point of lowest friction[48] and has been recently used with tribology to understand surface adsorption, particularly in boundary regime[8]. When comparing single microgels, each possesses significantly different hydrated masses ($p > 0.05$) positively correlating to elastic modulus (Fig. 4a) where viscoelasticity does not differ significantly (0.08–0.12, $p > 0.05$). When comparing frictional values to hydrated mass we see a negative correlation in the highest hydrated mass (PoPM10) and lower hydrated masses of other microgels (0.1 Pa m).

Despite PPM15 showing the most effective lubrication, interestingly hydrated mass and viscoelasticity (11.1 mg m$^{-2}$, $^{-\Delta D/\Delta f} = 0.08$) mirror data from that of native pea protein (11.1 mg m$^{-2}$, $^{-\Delta D/\Delta f} = 0.07$) when comparing with a previous study[8] (Table 2). However, striking differences are observed when applying a final buffer rinse, with increase in viscoelasticity (Supplementary Fig. 9), this suggests further hydration and swelling. Ultimately this may be key as to why microgels lubricate more effectively show less binding to the surface even if they support load (Table 1) in contrast to native proteins.

### Contribution to sustainability
With the drive to replace animal-based proteins with sustainable plant-based alternatives, the energy used to produce a refined protein powder (over 85% protein) from an original plant-based source must be considered, including the subsequent steps of fabricating these functional microgels. Life Cycle Analysis (LCA) have shown that the plant proteins produce much lower GHG emissions than meat proteins and such differences are mainly attributed to the primary production and plant-to-animal protein conversion (in-farm stages) rather than subsequent processing stages. It is now well-evidenced that in-farm processing produces the largest environmental impact[30,49,50] in contrast to beyond-farm processes. For instance, LCA of plant and meat burgers conducted by Heller et al.[51] demonstrated 90% less GHGs, 46% less energy, 93% less land and 99.5% less water when producing a plant-based burger using pea protein concentrate. These impressive improvements in climate markers are also observed for other analogues utilising soy protein isolates[52]. Although LCA of microgelation is beyond the scope of this work due to lab-scale production of this technology, LCA of other processing can be used as proxy to understand the environmental implications. Aganovic et al.[53] conducted a farm-to-gate analysis comparing thermal, pulsed electric field and high-pressure processing and reported that the environmental impact was overshadowed by the production of raw materials (20–64%) and packaging containers (85%). Hence, the additional processing of plant-based proteins towards fabricating these functional and lubricating microgels, which is in line with standard food processing practices appears to be negligible in terms of GHG emissions (with considerably lower energy requirements to those in the LCA of the thermally processing studied previously[53]). Of more importance, the high reward of improving plant protein lubrication performance via physical modification (without any chemical additives) offers a viable approach, (1) to create the next generation of sustainable, yet pleasurable, plant-based foods to enable the much needed large-scale transition from unsustainable animal protein-based diets and (2) reduce food wastage due to unpalatability, astringency challenges[9,24] and associated large scale consumer unacceptability of current plant-based foods[54].

## Discussion
Despite the immense effort to use sustainable plant proteins in food, their undesirable mouthfeel is a key bottle neck for consumer acceptability and hinders the transition to plant-based diets. Herein, we have presented a method to convert and optimise lubrication of plant proteins into effective lubricants by physically crosslinking them as microgels. Two commonly and commercially available plant proteins were used to prove this hypothesis, pea and potato, which were hydrated, gelled and homogenised into four types of microgel varying in concentration of protein and a range of volume fractions were tested. Through the use of DLS, AFM and rheology we physically observed these sub-micron sized microgels with low PDI values, suggesting a better control of size and stability over native proteins, latter being highly aggregated with limited functional performance. These microgels were extensively characterised in lubrication including hard-soft contact surfaces as well as 3D biomimetic tongue-like rough and wettable surfaces[45] to serve as effective proxies to sensory evaluation[6]. The microgel tribology performance was compared to the native protein, known to suffer with poor-lubrication and sensorially astringent properties, and also an O/W emulsion with desirable high-lubrication properties. Additionally, theoretical mechanisms of lubricity for each systems were determined using mathematical modelling, load-bearing experiments and surface properties measured using QCM-D.

Firstly, using size characterisation and rheology, we show that microgelation is an excellent technique to generate small, discrete, spherical, highly stable plant-based hydrated microgels with limited aggregation and mostly Newtonian behaviour as opposed by the shear-thinning behaviour and poorly stable native proteins that were self-aggregating. Plant microgels show tuneable viscosity modifying properties due to hydration and theorised swelling ability which can be controlled by altering protein type, crosslinking density and volume fraction.

Secondly, we evidence the excellent lubrication properties of microgels. PPM15, PoP5 and PPM7.5:PoPM5 achieved an order of magnitude decrease in friction from the native protein counterpart offering lubricity similar to O/W emulsion without the use of any lipids.

We further evidenced the lubrication improvement by comparing two tribology methodologies with use of additional biomimetic surfaces where similarities in friction responses were recorded, which provides strong in vitro *evidence* for mouthfeel performance. The topographic differences did relay importance as PoPM5, PoPM10 and PPM7.5:PoPM5 displayed reduced lubricity at low and high volume fractions, owing to an imposed papillae friction resulting in a macroscopic-aggregation hypothesis. This unexpected result depicts the importance of the use of alternate biomimetic surfaces for in vivo and in vitro comparisons, which must be standardised for future soft-tribology research. Nonetheless, understanding the real friction-mediated sensory analysis of plant protein microgels remains as a necessary undertaking, which is outside the scope of this study.

Finally, we unravel a mechanism of lubricity utilising adsorption, theoretical modelling and relative indentation. Adsorption measurements showed that viscoelasticity did not change from microgels but hydrated mass differed, relating to moduli and ability to take up water which provided a negative correlation to lubrication performance. However, a major difference was revealed in that subsequent washing of buffer resulted in little change in mass with evidence of swelling as viscoelasticity increased. Coupled with highly time and heat dependant microgel stability, this behaviour demonstrates the resistance of microgels to aggregation unlike native plant proteins. An alternative model for lubrication was determined, showcasing the similarity to O/W emulsion with dramatic lubricity differences converting native to micro-gelled protein owing to the effective height obtained by the swelling of microgels further complementing observation from adsorption mechanics. Further, from the relative indentation calculations we showcase the excellent loadbearing ability of microgels where water weeps out increasing localised viscosity contributing to hydration lubrication. It is postulated that the soft, hydrated microgels are able to flow and slide past one another as opposed to the aggregating-rubbing like feature of native plant proteins, reflected in viscosity, tribology, adsorption and theoretical results.

With growing sustainability needs, rise in vegetarianism, global protein malnourishment and inequity, the future world must look to plant based proteins in our growing population. Application of plants are at the forefront of food product development but currently limited by off-mouthfeel and poor functionality among a range of other barriers to adoption. Using a range of experimental and theoretical approaches, we confirm that microgelation is a viable technique to improve the lubricity, application and stability of plant protein in food. Ultimately, converting native plant proteins into microgels offers a facile platform to solve friction-related issues and combining this mechanistic work with sensory studies in the future will allow rapid transition from animal to palatable plant protein-based diets to promote planetary health.

## Methods
### Materials
Pea protein concentrate (PPC, Nutralys S85 XF) containing 85% protein was kindly gifted by Roquette (Lestrem, France). Potato protein isolate (PoPI) was purchased from Guzmán Gastronomía (Barcelona, Spain) containing 91% protein. Sunflower oil was purchased from a local supermarket (Morrison's) and used without further purification. HEPES (4-(2-hydroxyethyl)-1-piperazineethanesulfonic acid) were purchased from Fisher Scientific, UK. The solvent used was Milli-Q water (purified using Milli-Q apparatus, Millipore Corp., Bedford, MA, USA). Atomic force microscopy (AFM) cantilevers (HQ:CSC37/tipless/Cr–Au) were purchased from Windsor Scientific Ltd, UK. For creating the 3D-tongue like biomimetic surface with human tongue-like deformability, wettability and spatial distribution, size and shape of filiform and fungiform papillae, a similar procedure to that of ref. [45] was followed where Ecoflex 00–30 kit was purchased from Smooth-on Inc (Pennsylvania, U.S.A.) and the two components mixed in 1:1 w/w ratio with

the wettability modified by adding 0.5 wt% Span 80, purchased from Sigma-Aldrich (Dorset, U.K.). Mixing and degassing was performed using Thinky Planetary mixer system ARE-250, intertronics (Kidlington, U.K.) using a mixing cycle of 2 min at 2000 rpm followed by 1 min degassing at 2200 rpm. 3D printed tongue moulds were created from a Perfactory P3 mini 3D printer model (EnvisionTEC, Dearborn, U.S.A.) and used to cast a $2 \times 2\,cm$ 3D biomimetic tongue-like surface. All solutions were prepared from analytical grade chemicals unless otherwise mentioned.

### Preparation of plant protein microgels
Aqueous dispersions of plant protein microgels were prepared based on a similar procedure as described previously[55]. By thermal processing-induced disulphide crosslinking of proteins to form gels followed by shearing. Aqueous solutions of PPC (15.0 wt% total protein), PoPI (5.0 wt% total protein or 10.0 wt% total protein) and mixed PPC and PoPI (7.5 wt% and 5.0 wt% total protein, respectively) were prepared by dissolving the protein powders in 20 mM HEPES buffer at pH 7.0 for 2 h to ensure complete solubilisation. For samples undergoing microgelation, the aqueous protein dispersions were subsequently heated at 80 °C for 30 min and cooled in a cold water bath followed by storage at 4 °C overnight to form the thermally-crosslinked protein gels. These protein hydrogels were broken down using a hand blender (HB711M, Kenwood, UK) dispersed in HEPES buffer (70 vol%) for 5 min. These macrogel particle dispersions were degassed using Thinky Planetary mixer at 2200 rpm for 1 min. Protein dispersions were then passed through the PANDA homogeniser (Panda Plus 2000, GEA Niro Soavi Homogeneizador Parma, Italy) three times through a two-stage valve homogeniser operating at first/ second stage pressures of 250/ 50 bars. The resultant protein microgels are termed as PPM15, PoPM5, PoPM10, PPM7.5:PoPM5. Microgels were prepared multiple times to obtain at least three repeat measurement in analysis.

Volume fraction of microgels were calculated using Eq. 12:

$$vol\% = \frac{wt\%}{\rho} = \frac{x}{(x+y)\rho} \times 100\% \qquad (12)$$

where, $\rho$ is the density of the gel, $x$ is the weight of the gel and $y$ is the weight of the buffer. To obtain solutions of lower volume fractions (10–60 vol %), 70 vol% of the protein microgels were diluted with HEPES buffer at pH 7.0.

### Preparation of 20:80 oil in water emulsion
Potato protein (1.5 wt%) was dissolved in Milli-Q water with continuous mixing for 2 h. Subsequently, sunflower oil was added in a ratio of 20:80 w/w oil to aqueous phase containing protein. The mixture was subjected to mixing in the Ultra Turrax (Janke & Kunkel, IKA-Labortechnik) at 9400 rpm for 2 min, and then immediately passed (2 passes) through a two-stage valve homogeniser (Panda Plus 2000; GEA Niro Soavi Homogeneizador Parma, Italy) operating at pressures of 300 bars, respectively, droplet size distribution of the emulsion is provided in Supplementary Fig. 10. O/W emulsion were prepared multiple times to obtain at least three repeat measurement in analysis.

### Particle and droplet size and stability measurements
The mean hydrodynamic size ($d_H$) of the protein microgels were measured utilising dynamic light scattering (DLS) (Zetasizer Ultra, Malvern Instruments Ltd, Worcestershire, UK). The protein microgels were introduced into the Zetasizer in DTS0012 disposable cuvettes (PMMA, Brand Gmbh, Wertheim, Germany). The refractive index (RI) was set at 1.5 with an absorption of 0.001. The samples were equilibrated for 120 s and measured at 25 °C using non-invasive back-scattering technology at a detection angle of 173°. The particle size of the microgels were also measured as a function of storage time every

week for several months (data shown until 28 days) when stored at 22 °C and also when subjected to food processing (90 °C for 30 min). Droplet size distributions of emulsions were measured using static light scattering at 25 °C using Malvern Mastersizer 3000 (Malvern Instruments Ltd, Malvern, Worcestershire, UK). The mean particle size was reported as volume mean diameter ($d_{43}$). Means were calculated from six measurements on three independent samples.

## Atomic force microscopy

Dispersions of pea protein microgel (PPM15), potato protein microgel (PoPM5), potato protein microgel (PoPM10), and mixed pea and potato protein microgel (PPM7.5:PoPM5) were diluted by a factor of 1:50 v/v using 20 mM HEPES buffer solution at pH 7.0, deposited (100 μL) onto new, clean but untreated silicon wafers, and were allowed to adsorb for 10 min. Subsequently, to remove non-adsorbed microgels that could adhere onto the AFM tip, the solution was exchanged 5 times with 100 μl 20 mM HEPES buffer using a pipette while ensuring that the sample was kept constantly hydrated. Finally, the samples were transferred to an AFM for imaging. Topographic images were acquired using a Bruker Multimode 8 AFM equipped with a Bruker Nanoscope V controller. The plant protein microgels were imaged in contact mode, using silicon nitride AFM cantilevers (model MLCT-BIO-DC, Cantilever C) with a nominal spring constant of 0.01 N/m, purchased from Bruker AFM probes (Camarillo, CA). AC modes such as 8 kHz liquid tapping or 110 kHz FastScan-D tapping, or Peak Force tapping at 1–8 kHz that would preferably be used on delicate samples fail, probably due to instability and induced oscillations of the soft microgel material. The microgels are hugely sensitive to setpoint, with only a 50 pN window between lifting away from the surface, and disturbing/sweeping the microgels from the surface, necessitating use of the thermally stabilised MLCT-BIO-DC probe. Slow line rates of 0.5–0.8 Hz and gains at the very upper limits of stability were necessary to track the microgels with minimum disturbance. The measurements were performed at room temperature, using a fluid cell loaded with 20 mM HEPES buffer at pH 7.0. Images were acquired at 1024–1560 pixel resolution and processed using Bruker Nanoscope Analysis v3.0. Particle sizes from AFM images were measured using ImageJ-FIJI (NIH). After 2nd order flattening using masking, the images were 3 × 3 median filtered in Nanscope Analysis, then thresholded and converted to binary in ImageJ, and the binary image cleaned using the 'OPEN' function, which performs a dilate and erode operation. Finally, the particles were measured for area, position and length of the major and minor axis from a fitted ellipse. Particle diameters were calculated from the area, d = 2√(A/π). Diameters was also calculated from the average of major and minor axes, and the values found were virtually identical to the first method. Several thousand microgels were analysed for each sample to obtain robust statistical analysis of size and shape.

## Small deformation rheology

Viscoelasticity of the protein hydrogels was investigated by oscillatory shear rheology using modular compact rheometer (MCR-302 Anton paar, Austria) with a cone-and-plate geometry (CP50-1, diameter: 50 mm, cone angle: 1°) at 37 °C. The setup was initialized with a 0.208 mm gap between cone and plate with use of silicon oil around the cone geometry to prevent evaporation of the sample. Elastic ($G'$) and viscous ($G''$) moduli were measured by applying 0.01–10% strain at 1.0 Hz on the systems to determine linear viscoelastic region. A temperature ramp was performed to understand gelation characteristics and to gel the protein samples for frequency sweep, samples were heated (25–80 °C, rate of 0.08 °C/s) at a constant strain of 0.1% at 1 Hz and held at 80 °C for 30 min, the temperature was reduced to 37 °C and frequency sweep of 1–100 Hz at a strain of 0.1% initiated. Means were calculated from six repeat measurements. Flow curves of sheared protein hydrogels after passing through homogeniser i.e., the aqueous dispersions of the resultant protein microgels at a volume fraction of

10–70 vol% were measured at 37 °C using a stress-controlled rheometer (Paar Physica MCR 302, Anton Paar, Austria) equipped with a concentric cylinder geometry (inner diameter of the cup is 24.5 mm and diameter of the bob is 23 mm). Mean shear rates of 1 s⁻¹ to 1000 s⁻¹ were measured six times from three replicates of each sample.

## Large deformation rheology

Force distance curves of parent protein gels were measured using Texture Analyzer (TA-TX2, Table Micro Systems Ltd., Surrey, UK) attached with a 50 kg load cell. Samples were compressed using cylindrical probe (59 mm) at room temperature (22 °C) at a constant speed of 1 mm/s and deformation level set at 80% strain. A minimum of three replicates were measured for each parent plant protein gel sample in duplicate.

## Tribology

Tribological experiments were carried out using a tribology-cell attachment to the rheometer utilising steel ball (R = 7.35 mm) on three polydimethylsiloxane (PDMS) pins (6 mm height) inclined at 45° forming a steel ball-PDMS (hard-soft) contact. Samples were added in an enclosed chamber covering PDMS pins with steel ball geometry applying an evenly distributed load of 2.0 N. Upwards sliding speeds of 0.001–1.0 m/s were measured with pins remaining stationary generating three-sliding point contact. Measurements were performed at 37 °C with μ of HEPES measured as control. PDMS pins were cleaned using isopropanol then sonication in detergent for 10 min. Pins were replaced following signs of surface wear. A minimum of 6 measurements were carried out for triplicate samples. Load dependency experiments were performed on pea protein microgels (volume fraction = 70 vol%), native pea protein and O/W emulsion using same protocol using increasing loads ranging from 2.0 to 10.0 N.

In addition to the conventional tribology using hard-soft contacts surface, tribology was also performed using 3D biomimetic tongue-like similar to previous methodology[45] using a Kinexus Ultra+ rotational rheometer (Malvern Instruments, Malvern U.K.) equipped with 50 mm diameter stainless steel plate-on-plate geometry. The tongue-like soft, rough and hydrophilic surface was glued at the rim of the top plate forming a steel-elastomer contact. Experiments were performed in normal force ($F_N$) control and shear rate control mode. Normal force was 1.0 N with shear rates ranging from 0.005 to 1.0 s⁻¹. Friction coefficient μ was calculated using torque values following Eq. 13:

$$\mu = \frac{M}{RF_N} \tag{13}$$

where, $R$ is the plate radius ($R = 0.025$ m). Friction coefficient ($\mu$) is presented as a function of the linear speed $V_R$ at the rim of the plate and is calculated by $V_R = \Omega R$, where $\Omega$ is the angular speed.

## Adsorption

Adsorption behaviour of plant protein microgel were measured using quartz crystal microbalance with dissipation monitoring (QCM-D, E4 system, Q-Sense, Biolin Scientific, Sweden). Using a similar procedure to ref. [7], silicon sensors coated with PDMS were prepared by spin-coating (QSX-303, Q-Sense, Biolin Scientific, Sweden) with a solution of 0.5 wt% PDMS in toluene at 5000 rpm for 30 s with an acceleration of 2500 rpm/s, before leaving overnight in a vacuum oven at 80 °C. Before use, PDMS-coated crystals were further cleaned by immersing in toluene for 1 min, then 1 min in isopropanol and a final immersion in Milli-Q for 5 min before being dried using nitrogen gas.

Protein solutions were made at a volume fraction of 1 vol% and were equilibrated in buffer at (25 °C) before measurement. The flow rate was controlled using peristaltic pump at a rate of 100 μL/min at 25 °C. HEPES buffer solution was initially injected to obtain a stable baseline reading and then the prepared protein solutions were injected until

equilibrium adsorption i.e., no change in frequency ($f$) or dissipation ($D$) was recorded. Finally, the buffer was used once more to remove any non-adsorbed protein microgels. Hydrated mass was calculated from the frequency data using viscoelastic Voigt's model[56] using Smartfit Model by Dfind (Q-Sense, Biolin Scientific, Sweden) software. The 3rd, 5th, 7th and 11th overtones were taken into account for data analysis and only 5th overtone is shown in the results. A minimum of three replicates were measured for each protein sample in duplicate.

## Statistical analysis

All results are reported as means and standard deviations on at least three measurements carried out on three independent samples prepared on separate days. Statistical analysis on the significance between data sets was calculated using analysis of variance (ANOVA) with Tukey post hoc test, significance level $p < 0.05$. All model calculations were performed using R version 4.1.0 (2021-05-18), model parameters were estimated using non-linear solvers in the packages nlstools, nls2 and minpack.

## Reporting summary

Further information on research design is available in the Nature Portfolio Reporting Summary linked to this article.

## Data availability

Source data that support the plots within this paper and other findings are provided with this paper at https://doi.org/10.6084/m9.figshare.22722718 Source data are provided with this paper.

## Code availability

Source code that supports the numerical modelling within this paper is provided with this paper.

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

## Acknowledgements

Funding from the European Research Council (ERC) under the European Union's Horizon 2020 research and innovation programme (grant agreement n° 757993) is acknowledged for this work. A.S. also acknowledges UKRI Horizon Europe Guarantee Fund (EP/X03514X/1). D.D. would like to acknowledge the support received *via* his Royal Academy of Engineering Research Chair in Complex Engineering Interfaces (RCSRF2122-14-143).

## Author contributions

A.S. conceived the concept of the study and supervised the study. B.K. designed the experimental protocol with inputs from A.S. and D.D., designed the microgels, carried out all the experiments and prepared the initial draft of the manuscript. E.L. and S.D.C. designed and carried out the atomic force microscopy experiments. M.H. and R.E. developed the theoretical framework and M.H. performed the numerical calculations. B.K. revised the paper with contributions from A.S., M.H., E.L., R.E., S.D.C. and D.D.

## Competing interests

The authors declare no competing interests.
