## [Peer Review File · Nature Communications]

Transforming sustainable plant proteins into high performance lubricating microgelsReviewers' Comments:

Reviewer #1:

Remarks to the Author:

The manuscript entitled "Transforming sustainable plant proteins into high performance food lubricants" reported that the formulated microgels from plant proteins can improve their lubricity remarkably by both experiments and numerical modeling. The authors took the pea, potato proteins and their mixtures as examples. They found that the plant proteins can be dispersed well in the microgels and likely reduce the undesirable mouthfeel, as a potential fat mimetic. These findings are very interesting, and I would like to recommend it to be published if the following comments are addressed well.

Abstract

Line 27-32: The author should clearly mention why plant based microgels can be potentially used to improve lubricity and reduce fractions significantly. Please give more introduction on the rationale.

Introduction:

Line 52-54: "In order to address ... food design" should cite the reference related.

Line 69-73: Which kind of protein has better lubrication performance, animal, or plant? Please introduce more here because I'm curious about the performance of animal proteins (such as whey protein). Can animal-based microgels improve lubrication performance as well?

Line 77: What does the "first time" refer to? the plant protein or the microgels?

Results:

Fig1. Please correct "PoPM15" as "PoPM10", and "PoPI5" as "PoPM5"

What is the stability of microgels? Modern food processing usually includes a higher temperature or sterilization (~93 °C). The native protein will be denatured around 60 to 90 °C. Given that the protein microgels have already got denatured and gelled, will this resistance to this higher temperature? The author should discuss the potential impact of temperature.

The performance of PoPM5 and PoPM10 are significantly different (Fig 3). Please provide more discussion to explain the concentration-dependent effect.

Others:

Please give us the points of using pea and potato proteins as representative examples. Are these observations applicable to other plant proteins? If so, please give your rationale.

In this paper, the authors mentioned the sustainability is very important, however, from the Methods section, we can see there are lots of energy inputs during the preparation of plant protein microgels. The authors should explain why the plant protein microgels are still more sustainable after many times of pass by the high-energy top-down methods (homogenizers and mixers). Or are there better ways to fabricate the microgels with more energy-efficient methods? If so, please clarify the future improvements.

The authors mentioned the native proteins. However, it is usually hard to extract pure native proteins. Because during protein fraction and extraction, protein structure is easily denatured and unfolded. Also, the PDI data also show that these "native proteins" are highly aggregated. Are these consistent with the observations from other studies?

Reviewer #2:

Remarks to the Author:

The manuscript contains an interesting topic, and shows the lubrication properties of different plant proteins. Although some parts are interesting, the paper contains a lot of hypotheses and speculation, and misses a clear systematic approach to provide proof for such hypotheses. In addition, the protein particles are poorly defined. The authors provide information on the size, but it is known that other aspects as shape and elasticity (deformation) are also very relevant. Information on the native proteins is missing, but is important to understand the results. Commercial plant proteins are often already aggregated (and could thus be considered a microgel by themselves). So what are the properties of the native protein? Is it all soluble? Or does it consist out of microgels?

Furthermore, the authors mention a few times in their manuscript (abstract, conclusion) the terms astringency, mouthfeel, etc. However, they do not provide anything on these characteristics. They only focus on the lubrication properties. So they cannot really claim those statements. They also highlight a lot of their own papers, but forget to mention other literature in which opposite results are sometimes shown. (for example: whey protein microgels have also been shown to give high friction, whey protein particles do not always show shear thinning behavior. At high concentrations, they have also been shown to give shear thickening behavior..). Figure 4 seems to be the most important results, but the symbols are so large that it is difficult to separate the results. The results are also not discussed and explained well. The results of the emulsion seems rather weird. Even though they mention that this has been shown before, this is not a normal behavior. Even with coalescence, emulsions do not show this behavior. Although the mathematical model is a nice addition, they do not really couple equation 11 to the results? What information do they get from it? And why only show this for two of the samples?

Overall, the manuscripts contains a high level of speculation of the discussions without a systematic experimental approach to proof some of these aspects. Too many different hypothesis have been introduced without clear results. Therefore, I believe the level is not sufficient enough for a high impact journal as Nature communications.

Reviewer #3:

Remarks to the Author:

Without a doubt the authors have a strong expertise in the field of food colloids, rheological and tribological characterisation of foods and numerical methods. As a consequence I highly appreciate the experimental approach in the present study. The theoretical considerations on lubrication performance are a significant contribution to the field, which has not been published before. However, the innovative character of plant protein microgel utilisation should be elaborated in more detail. Tribological characterisation of plant proteins as well as microgel characterisation of dairy based microgels has already been published by the authors. Plant protein gelation, especially for pea protein, is also well described in several publications. So the reason, why findings of the present study are not on hand from these investigations and the knowledge gap should be elaborated in more detail. I agree that literature on plant protein microgel dispersions is scarce, but authors should explain the scientific innovation justifying publication in Nature Communications in more detail.

Apart from some minor issues listed below I have some more fundamental points, which should be addressed:

Motivation for the setup is not entirely clear to me. Why did the authors decide for a blend of the two proteins and why did the authors vary the protein content, if they want to compare the plant proteins. If proteins have a quite large difference in denaturation temperature and the do not complex in one particle (line 119f), what is the expected benefit of the binary blend and the oscillatory shear rheology experiments of the blend? In line 189 you state that the microgels are sub-micrometric units of the parent hydrogels. This is somehow contradictory to the statement previously cited.

Furthermore, the authors should explain in more detail, why they chose to compare the microgel particles to native proteins. With a high percentage of aggregated and insoluble protein, conclusions like in line 155 ("rheological behaviour of microgels is in sharp contrast to native proteins") appear obvious and the shear-thinning behaviour of plant proteins has been reported before. Also the difference lubrication as discussed in line 230ff. is an obvious consequence of the gelation.

Line 47, Typo: "it" instead of "it"

Line 49, Typo: "dairy" instead of "diary"

Line 52: I do not agree, that lubrication issues in plant-based foods are addressed through ultra-processing and result in unsustainable, unhealthy and undesirable foods. I am convinced that the authors would also not agree to such a generic statement and recommend to either give specific examples or re-phrase. Personally I would completely remove this aspect, since the materials science aspect of the poor lubrication is sufficient to justify the research.

Line 96: I recommend to add an explanation, why the percentage of protein has been varied when using different proteins.

Line 115, Figure title: one may misunderstand it in a way that the protein content of the ppc has been 15%

Line 116: Add "particles" after "Individual microgels"

Line 125: What is the reason for the variation in DLS. Later on in the discussion authors should comment on the impact of the size on DLS analysis. Is the motion of the particles analysed in DLS exclusively based on electrophoretic motion?

Line 144: Please explain what you mean by spreading-like behaviour and which result you are exactly referring to.

Line 195: What does "stark" mean? I am sorry, but I have never read this word before in a manuscript.

Line 218. I assume that drop size in the emulsion plays a major role in tribology. How was it adjusted to microgel size. At least I recommend to report the drop size distribution.

Figure 4: too crowded for me, it is not possible to properly compare the different samples.

Furthermore I would recommend to indicate the different lubrication regimes. In the presentation comparison is done at a specific volume fraction. MAYbe the whole figure should be adapted in that way, so that the figure matches the story line in the text.

Line 453: "resulted increases" - structure of the sentence not clear

Line 455: please discuss in more detail the nature of the aggregates

Line 513: With an oil load of 20% the emulsion is not a high fat emulsion in my opinion.

Line 582ff. : Why did the authors use two different homogenisers? This will inevitably lead to differences in microgel particle structure as well as rheological and tribological behaviour.

Line 586: Please decide, if you name the final microgel, particle or microgel particle and consequently use the term throughout the text.

Line 600/631/675: It is not clear whether multiple preparations of the microgel systems have been performed. means and standard deviations should only be provided, if they are based on true replicates rather than analytical replicates.

Line 625: During gel preparation the time was 30min, in rheology the authors reduced to 10 min. This may heavily affect gel structure and hinders a comparison.

Reviewer #1 (Remarks to the Author):

The manuscript entitled “Transforming sustainable plant proteins into high performance food lubricants” reported that the formulated microgels from plant proteins can improve their lubricity remarkably by both experiments and numerical modelling. The authors took the pea, potato proteins and their mixtures as examples. They found that the plant proteins can be dispersed well in the microgels and likely reduce the undesirable mouthfeel, as a potential fat mimetic. These findings are very interesting, and I would like to recommend it to be published if the following comments are addressed well.

****Response:** We thank the reviewer for their valuable time and efforts in positively evaluating the manuscript and recommending publication. We have done substantial revision in the manuscript to address all the points raised by the reviewer.

Abstract

Line 27-32: The author should clearly mention why plant based microgels can be potentially used to improve lubricity and reduce fractions significantly. Please give more introduction on the rationale.

****Response:** We thank the reviewer for this suggestion. We have also now included this new information in the introduction section of the revised manuscript on issues with plant proteins and benefits from microgelation.

Lines 47-56 in the revised manuscript now read as “It is now well-evidenced that one of the primary barriers to adoption of plant protein is their negative “astringent” *i.e.* dry, puckering, non-juicy perception. Oral lubrication has shown to be crucial to understand, and serves as a well-acknowledged *in vitro* proxy for quantifying friction-related mouthfeel characteristics¹. Multiscale tribology measurements across laboratories supported by sensory trials have revealed that plant proteins increased oral friction due to particle-like protein-protein aggregation and jamming as well as interaction with saliva, in contrast to dairy proteins²⁻⁶. Such lubrication failure or “delubrication”⁴ highlights a major issue if plant proteins are to be used instead of conventional animal proteins, incurring adverse textural modifications and development of these astringent, drying characteristics would reduce consumer acceptability of plant protein-rich foods.”

Lines 77-84 in the revised manuscript now read as “Additionally, plant proteins also face intra-variability within the protein type, where both natural climate conditions and industrial processing can result in non-standardised proteins with a range of poor solubility, limited functionality and poor hydrating ability^{7,8} resulting in oral dryness. These present major challenges for plant proteins to be used as a food ingredient where microgelation can be a novel structuring platform to standardize and overcome these oral frictional hurdles, which is the key question this study answers. Therefore, lubrication evidences on microgelation of plant proteins are imperative before such microgels can be applied as a highly functional, pleasurable ingredient for designing next generated plant protein-reformulated foods⁹.”

In addition, we included a **new schematic Figure 1** detailing the experimental set up and structural difference of microgels versus native proteins underpinning the benefits in lubrication performance in the microgels.

Fig 1| Schematic illustration of microgelation of native plant proteins. Visual representation of microgelation procedure. Native plant proteins are highly aggregated causing functional and sensory problems in food design. By hydrating them with water and thermally gelling using hydrophobic interactions, hydrogen bonding and disulphide-based covalent crosslinking occurring without any added crosslinking agents, the native plant proteins act as connecting particulate points in a highly percolating hydrogel network, which is then converted into gel-like particles *via* controlled homogenization consisting of 5-15 wt% protein and 85 – 95 wt% water. These microgels remove functional issues associated with native protein allowing for improved functional application of plant proteins in food.

Introduction:

Line 52-54: “In order to address ... food design” should cite the reference related.

****Response.** We revisited this statement in the original submission and found it to be generic and have therefore removed this in the revised manuscript.

Line 69-73: Which kind of protein has better lubrication performance, animal, or plant? Please introduce more here because I’m curious about the performance of animal proteins (such as whey protein). Can animal-based microgels improve lubrication performance as well?

****Response.** In general, animal proteins have better lubrication performance than plant proteins. This has been evidenced both by *in vitro*^{2,3} and *human sensory trials*⁴ and highlighted in in the introduction.

Lines 50-53 in the revised manuscript state that “Multiscale tribology measurements across laboratories supported by sensory trials have revealed that plant proteins increased oral friction due to particle-like protein-protein aggregation and jamming as well as interaction with saliva, in contrast to dairy proteins²⁻⁶.”

There are studies on microgel formulated using dairy proteins showing lubrication benefits. This is now highlighted in **Lines 68-72** which read as “Animal protein-based, polysaccharide as well as synthetic microgels such as whey protein-based, polyacrylic acid-based and carrageenan-based microgels have demonstrated varying degree of lubricity¹⁰⁻¹² depending upon volume fraction and elasticity of particles entrained in the contact. Often such

lubrication mechanism has been governed by viscosity modification with microgels acting as physical surface separators^{11,12} or by so-called (often debated) ball bearing mechanisms^{13,14}.”

Line 77: What does the “first time” refer to? the plant protein or the microgels?

****Response.** “First time” refers to the fact that it is the first evidence using theoretical and experimental approaches that plant-based microgels designed using lubricating plant proteins show outstanding lubrication similar to those in fat-based emulsions We have rephrased this in the revised manuscript in **Lines 85-87**, which read as “Herein, we demonstrate using a combination of experimental and theoretical approaches that engineering physically cross-linked, soft, sub-micron sized plant-based microgels offers superior lubrication performance as compared to their parent native protein counterparts for the first time.”

Results:

Fig1. Please correct “PoPM15” as “PoPM10”, and “PoPI5” as “PoPM5”

****Response.** We thank the reviewer for pointing this out, this mistake is now corrected in the revised manuscript (now **Figure 2**).

What is the stability of microgels? Modern food processing usually includes a higher temperature or sterilization (~93 °C). The native protein will be denatured around 60 to 90 °C. Given that the protein microgels have already got denatured and gelled, will this resistance to this higher temperature? The author should discuss the potential impact of temperature.

****Response.** We thank the reviewer for asking this question. Indeed the protein is denatured and gelled and this gives further thermal resistance of these structured, microgelled proteins. To support further, we have now conducted two additional sets of experiments. The microgel has shown strong resistance to change in hydrodynamic size (measured using dynamic light scattering) when subjected to heating (90 °C for 30 minutes) (see **new Supplementary Table S2**) as well as storage time (see **new Supplementary Table S1**).

We also conducted further sedimentation experiments showcasing a native protein vs microgel (**new Supplementary Figure S1**). Within few hours native proteins show sedimentation in contrast to a microgel which even after a month of storage shows no sedimentation.

Lines 160-165 in the revised manuscript now read as “For instance, microgels even after a month of storage show no sedimentation in comparison to the non-microgelled counterparts that sediment within few hours (Supplementary Fig. S1) where there is little change in particle size (Supplementary Table S1). Even when the microgels were further processed such as those simulating food processing (*e.g.* thermally treated at 90 °C for 30 min), no marked change was observed in hydrodynamic diameter or polydispersity index (Supplementary Table S2) highlighting the excellent thermodynamic stability of these microgels.”

The performance of PoPM5 and PoPM10 are significantly different (Fig 3). Please provide more discussion to explain the concentration-dependent effect.

****Response:** We link this to the difference in deformation of the microgels where we have now carried out additional large deformation experiments on the parent gels (see new **Figures 4a** and **4b**) that demonstrates that explains that softer microgels were able to deform more easily than the stiffer harder microgels.

This is now stated in **Lines 268-270** in the revised manuscript, which read as “PoPM10 packing limit is met earlier, most likely due to the high content of soluble protein (100%~ solubility at pH 7³) and higher modulus (see Figs. 4a, 4b) allowing for the superior viscosities recorded.”

Fig 4| Rheological properties of the parent plant protein gels and volume fraction-dependent apparent viscosities of the microgels. Storage modulus (a) and Young’s modulus (b) of parent plant protein gels with apparent viscosities (η) of microgels prepared using pea protein concentrate to form a 15.0 wt% total protein microgel, (PPM15), potato protein isolate to form a 5.0 wt% total protein microgel (PoPM5), potato protein isolate to form a 10.0 wt% total protein microgel, (PoPM10), and using a mixture of pea protein concentrate at 7.5 wt% total protein and potato protein isolate at 5.0 wt% total protein microgel (PPM7.5:PoPM5) with corresponding storage modulus (G') of parent plant protein gels (a) as a function of volume fractions (ϕ) at pH 7.0 at shear rates of (c) 0.1 s^{-1} and (d) 50 s^{-1} , the latter representing orally relevant shear rates performed at $37 \text{ }^\circ\text{C}$. Data was recorded with shear rate increasing from 0.1 to 50 s^{-1} , representing an average of six measurements on triplicate samples ($n = 6 \times 3$) with error bars representing standard deviations. Different lower-case letters in the same bar chart indicate a statistically significant difference ($p < 0.05$). The original temperature ramp and frequency sweeps of the parent heat set gelled proteins are shown in **Fig. S3** and **Fig. S4**, respectively. The true stress-strain curves are shown in **Fig. S5** from which the Young’s moduli are computed. Original flow curves for the microgel dispersions at each volume fractions are shown in **Fig. S6**.

Others:

Please give us the points of using pea and potato proteins as representative examples. Are these observations applicable to other plant proteins? If so, please give your rationale.

****Response.** In general plant proteins generally contain globular forms of multimer storage proteins. Pea protein is limited in aqueous solubility and results in high friction and sensorial astringency. Potato proteins are composed of rather unique glycoprotein *i.e.* patatins. With comparatively higher fractions of soluble protein but still suffer from astringency issues due to high surface hydrophobicity. Hence we used these two distinct types of plant proteins and demonstrated that microgelation can be effective for both of these two fundamentally different proteins and thus can be applied to other plant proteins for achieving lubrication benefits.

Lines 89-97 in the revised manuscript now read as “This study utilised pea protein and potato protein as typical exemplar plant proteins which take the form of globular multimer storage proteins. Pea proteins are composed mostly of 11S and 7S globulins¹⁵ and frequently reported for being of limited aqueous solubility^{3,8,15} result in high friction and sensorial astringency^{4,16}. On the other hand, potato proteins are composed of rather unique, globular, glycoprotein *i.e.* patatins¹⁷, with comparatively higher fractions of soluble protein^{3,17,18} but still suffer from astringency issues^{3,16} due to high surface hydrophobicity¹⁹. A blend of the proteins were also investigated to determine any synergistic or detrimental effect on lubricity. The latter approach also provides a way to enhance the amino acid profile of possible formulations through protein complementation, a crucial consideration in non-animal protein diets²⁰.”

In this paper, the authors mentioned the sustainability is very important, however, from the Methods section, we can see there are lots of energy inputs during the preparation of plant protein microgels. The authors should explain why the plant protein microgels are still more sustainable after many times of pass by the high-energy top-down methods (homogenizers and mixers). Or are there better ways to fabricate the microgels with more energy-efficient methods? If so, please clarify the future improvements.

****Response.** We thank the reviewer for raising this important point. It is now well-evidenced that the differences between plant and animal proteins in contribution to greenhouse gas emissions (GHG) is mainly linked to primary production and additional processing such as heat treatment (post farm-gate) adds negligible contribution as compared to the primary production. Although a full life cycle assessment (LCA) of microgelation is beyond the scope of this work as it is not reasonable to conduct LCA on lab-scale fabrication but we have summarized this in the revised manuscript supported by relevant proxies reported in literature.

Lines 556-579 in the revised manuscript read as “**Contribution to sustainability.** With drive to replace animal-based proteins with sustainable plant-based alternatives, the energy used to produce a refined protein powder (over 85% protein) from an original plant-based source must be considered, including the subsequent steps of fabricating these functional microgels. Life Cycle Analysis (LCA) have shown that the plant proteins produce much lower GHG emissions than meat proteins and such differences mainly is attributed to the primary production and plant-to-animal protein conversion (in-farm stages) rather than subsequent

processing stages. It is now well-evidenced that in-farm processing produces the largest environmental impact²¹⁻²³ in contrast to beyond-farm processes. For instance, LCA of plant and meat burgers conducted by Heller et al²⁴ demonstrated 90% less GHGs, 46% less energy, 93% less land and 99.5% less water when producing a plant-based burger using pea protein concentrate. These impressive improvements in climate markers are also observed for other analogues utilising soy protein isolates²⁵. Although LCA of microgelation is beyond the scope of this work due to lab-scale production of this technology, LCA of other processing can be used as proxy to understand the environmental implications. Aganovic et al.²⁶ conducted a “farm to gate” analysis comparing thermal, pulsed electric field and high-pressure processing and reported that the environmental impact was overshadowed by the production of raw materials (20-64%) and packaging containers (85%). Hence, the additional processing of plant-based proteins towards fabricating these functional and lubricating microgels which is in line with standard food processing practices appears to be negligible in terms of GHG emissions (with considerably lower energy requirements to those in the LCA of the thermally processing studied previously²⁶). Of more importance, the high reward of improving plant protein lubrication performance *via* physical modification (without any chemical additives) offers a viable approach, 1) to create next generation of sustainable, *yet pleasurable*, plant-based foods to enable the much needed large-scale transition from unsustainable animal protein-based diets and 2) reduce food wastage due to unpalatability, astringency challenges^{4,16} and associated large scale consumer unacceptability of current plant-based foods²⁷.”

The authors mentioned the native proteins. However, it is usually hard to extract pure native proteins. Because during protein fraction and extraction, protein structure is easily denatured and unfolded. Also, the PDI data also show that these “native proteins” are highly aggregated. Are these consistent with the observations from other studies?

****Response.** Indeed reviewer is correct, we used the term “native protein” to highlight it is the commercially available concentrate without microgelation. We now show the structure in Fig 1 as mentioned previously and have referenced a number of papers that have used dynamic light scattering showing the high level of aggregation and polydispersity in these native plant proteins *i.e.* non-microgelled plant proteins within this study.

We have now added **Lines 159-161** supported by references which read as “When comparing to native proteins *i.e.* non-microgelled plant protein isolates/ concentrates, these often produce more variation in size as they are highly polydisperse, aggregated and may need filtering due to sedimentation^{3,28}.”

Reviewer #2 (Remarks to the Author):

The manuscript contains an interesting topic, and shows the lubrication properties of different plant proteins. Although some parts are interesting, the paper contains a lot of hypotheses and speculation, and misses a clear systematic approach to provide proof for such hypotheses. In addition, the protein particles are poorly defined. The authors provide information on the size, but it is known that other aspects as shape and elasticity (deformation) are also very relevant.

****Response.** We thank the reviewer for taking the time to evaluate the manuscript. We agree elasticity and shape can have important contribution to lubrication performance. We have now included **new Fig. 2, Supplementary Fig. S2, Fig. 4b, Supplementary Fig. S5** providing new information on size, shape and elasticity of the microgels.

Elasticity of microgels. Although we had already provided information on elasticity in original submission which has been not only been used in the tribological discussion but also in the numerical modelling supporting the experimental results, we have now further conducted large deformation experiments and therefore included further information on Young's modulus of the parent gels in **new Fig. 4b** (see **new Supplementary Fig. S5** for the true stress-strain curves). Hence, the new information on elasticity is complete and provides robustness to the characterization of microgels.

Fig 4| Rheological properties of the parent plant protein gels and volume fraction-dependent apparent viscosities of the microgels. Storage modulus (a) and Young's modulus (b) of parent plant protein gels with apparent viscosities (η) of microgels prepared using pea protein concentrate to form a 15.0 wt% total protein microgel, (PPM15), potato protein isolate to form a 5.0 wt% total protein microgel (PoPM5), potato protein isolate to form a 10.0 wt% total protein microgel, (PoPM10), and using a mixture of pea protein concentrate at 7.5 wt% total protein and potato protein isolate at 5.0 wt% total protein microgel (PPM7.5:PoPM5) with corresponding storage modulus (G') of parent plant protein gels (a) as a function of volume fractions (ϕ) at pH 7.0 at shear rates of (c) 0.1 s^{-1} and (d) 50 s^{-1} , the latter representing orally relevant shear rates performed at $37 \text{ }^\circ\text{C}$. Data was recorded with shear

rate increasing from 0.1 to 50 s⁻¹, representing an average of six measurements on triplicate samples (n = 6 × 3) with error bars representing standard deviations. Different lower-case letters in the same bar chart indicate a statistically significant difference (($p < 0.05$). The original temperature ramp and frequency sweeps of the parent heat set gelled proteins are shown in **Fig. S3** and **Fig. S4**, respectively. The true stress-strain curves are shown in **Fig. S5** from which the Young's moduli are computed. Original flow curves for the microgel dispersions at each volume fractions are shown in **Fig. S6**.

Lines 201-212 in the revised manuscript now read as “Before characterising viscosity, it is nevertheless important to understand the stiffness of the microgels which may influence their viscous dissipation such that higher storage modulus (G') of microgels corresponding to higher viscosities of the microgel dispersion¹². We assume microgels are sub-micrometric units of the parent hydrogels and thus possess the same elasticity. To quantify this, oscillatory shear rheology (see temperature ramp and frequency sweeps in Supplementary Figs. S3 and S4, respectively) was performed on the parent protein hydrogels prior to shearing to obtain G' and loss (G'') modulus (Fig. 4a) and large scale deformation tests were performed (Supplementary Fig. S5) to calculate the Young's modulus (Fig. 4b). Typically, the higher the protein content of the microgel, the higher the G' . For instance, G' of the parent gels for PoPI5 (~800 Pa), was an order of magnitude lower than that of PoPI10 (~8500 Pa), explaining the value of PoPM10 viscosity observed in Fig. 4c,d compared to the softer, easily compressible microgels contributing to lower viscosity, which is true of PoPM5.”

Shape of microgels. We have now included more precise AFM imaging (**revised Fig. 3**) as well as conducted additional shape analysis using AFM (**Supplementary Fig. S2**) with statistical image analysis of > 1000 microgel particles. Here, we can clearly evidence that these microgels were between spherical (1:1 major: minor axis) to flattened pan-cake shaped (2:1 major: minor axis).

Lines 168-198 in the revised manuscript now read as “To investigate microgel morphology in more detail, atomic force microscopy (AFM) was employed to image fully hydrated microgels, shown in Fig 3 with corresponding particle size distributions. All microgels are in the same general size range of 50-200 nm as found by DLS. (Fig. 2b).

The mixed PPM:PoPM (Fig 3d) system displays two distinctly sized population of microgels which could be the result of sub-unit protein complexes forming from pea and potato protein as discussed previously in DLS. PoPM5 (Fig 3b) is remarkably similar in size in both techniques, 67 nm (DLS) vs 73 nm (AFM). In the other samples the AFM size was slightly smaller than DLS; for PoPM10 (Fig 3c) the diameters DLS:AFM were 132 nm:109 nm, and for the mixed PPM:PoPM were 90 nm: 78 nm for the first peak, and 260 nm: 158 nm for the second. This reduction in size can be explained by noting that the loosely structured brush-like features extends from the surface of microgels and occupies a larger hydrodynamic volume, we speculate the increased overestimation from DLS could be related to the hydration shell whilst force microscopy measures the protein core, this explains the similar size of PoPM5 and larger size of PoPM10 as more protein have an extended influence on hydrodynamic diameter. PPM15 (Fig 3a) showed a larger discrepancy, 204 nm (DLS) vs 79 nm (AFM). This could be explained by the presence of a small number of polydisperse large particles or aggregates in this sample which could skew the DLS measurement to a higher value (these aggregates were not seen in the distributions of PoPM5 and PoPM10) or show the presence of a highly hydrated shell spanning far from the core. Shape analysis (Fig S2) showed the majority of particles were spherical or near spherical with an aspect ratio < 2:1, although with increasing aspect ratio with size, which is probably explained by the larger

particles being aggregates. Overall microgels take on a smooth and spherical shape showing convex spreading on the surface (Fig 3a-d) which has also been observed for synthetic and whey protein microgels in previous studies^{12,29}.”

Fig 3 | Images of plant protein microgels on silicon under buffer. Topographic images and respective histograms showing diameters of aqueous dispersions of protein microgels prepared using (a) pea protein concentrate to form a 15.0 wt% total protein microgel, (PPM15), potato protein isolate to form a 5.0 wt% total protein microgel (PoPM5), potato protein isolate to form a 10.0 wt% total protein microgel, (PoPM10), and using a mixture of pea protein concentrate at 7.5 wt% total protein and potato protein isolate at 5.0 wt% total protein microgel (PPM7.5:PoPM5) obtained by atomic force microscopy (AFM). Most of the microgels are distributed homogeneously on the surface.

Supplementary Fig. S2 | Shape analysis of all particles measured in the microgel particle distributions shown in Fig 3, several thousand particles per sample. Ellipses were fitted to each particle, and the graph plots the short or minor axis vs the long or major axis. Hence, the perfectly spherical particles will follow the red line with a 1:1 aspect ratio. Most particles in all samples were between spherical and a 2:1 aspect ratio represented by the green line. A general trend found is the increase in aspect ratio as particle size increases, represented by the dashed red fit lines (note: the Michaelis-Menten fit equation used is not physically relevant). This could be explained by the larger particles being randomly shaped aggregates of the smaller particles.

Information on the native proteins is missing, but is important to understand the results. Commercial plant proteins are often already aggregated (and could thus be considered a microgel by themselves). So what are the properties of the native protein? Is it all soluble? Or does it consist out of microgels?

****Response.** Indeed “native plant proteins” are highly aggregated. We have now added **Lines 159-161** supported by references in this regard, which read as “When comparing to native proteins *i.e.* non-microgelled plant protein isolates/ concentrates, these often produce more variation in size as they are highly polydisperse, aggregated and may need filtering due to sedimentation^{3,28}.”

We would like to pinpoint that microgels are very different from the aggregated native proteins. To clarify, we have included a **new schematic Figure 1** in the revised manuscript detailing the experimental set up and structural difference of microgels versus native proteins underpinning the benefits in lubrication performance in the microgels.

Fig 1| Schematic illustration of microgelation of native plant proteins. Visual representation of microgelation procedure. Native plant proteins are highly aggregated causing functional and sensory problems in food design. By hydrating them with water and thermally gelling using hydrophobic interactions, hydrogen bonding and disulphide-based covalent crosslinking occurring without any added crosslinking agents, the native plant proteins act as connecting particulate points in a highly percolating hydrogel network, which is then converted into gel-like particles *via* controlled homogenization consisting of 5-15 wt% protein and 85 – 95 wt% water. These microgels remove functional issues associated with native protein allowing for improved functional application of plant proteins in food.

Noteworthy, a microgel is a highly hydrated 85-95% water containing percolating gel network whilst native proteins are particle aggregates. We have highlighted this in **Lines 107-110** in the revised manuscript, which read as “Four types of microgels were fabricated utilising a top-down methodology involving thermal gelation-induced physical crosslinking of the *otherwise sedimenting* (Supplementary Fig. S1) plant proteins resulting in a percolating, viscoelastic protein-based hydrogels followed by controlled shearing into microgels (Fig. 1, see detailed preparation in method section).”

The fundamental information on native protein structure/size/solubility has been extensively characterised in previous literature. The properties of native proteins have been referenced or where relevant compared against microgels throughout the manuscript. Additionally we have conducted sedimentation experiments and sizing showing the stability of microgels over native proteins (see **Supplementary Fig. S1** and **Table 1**)

Lines 159-164 in the revised manuscript now read as “When comparing to native proteins *i.e.* non-microgelled plant protein isolates/ concentrates, these often produce more variation in size as they are highly polydisperse, aggregated and may need filtering due to sedimentation^{3,28}. For instance, microgels even after a month of storage show no sedimentation in comparison to the non-microgelled counterparts that sediment within few hours (Supplementary Fig. S1) where there is little change in particle size (Supplementary Table S1). For instance, microgels even after a month of storage show no sedimentation in comparison to the non-microgelled counterparts that sediment within few hours (Supplementary Fig. S1) where there is little change in particle size (Supplementary Table S1).”

Furthermore, the authors mention a few times in their manuscript (abstract, conclusion) the terms astringency, mouthfeel, etc. However, they do not provide anything on these characteristics. They only focus on the lubrication properties. So they cannot really claim those statements.

****Response.** We understand the reviewer's concern. There has been well-established literature on this already that plant proteins are astringent and suffer from delubrication. We have already stated this in original submission but highlighted this further in the revised manuscript in **Lines 47-55**, which read as "It is now well-evidenced that one of the primary barriers to adoption of plant protein is their negative "astringent" *i.e.* dry, puckering, non-juicy perception. Oral lubrication has shown to be crucial to understand, and serves as a well-acknowledged *in vitro* proxy for quantifying friction-related mouthfeel characteristics¹. Multiscale tribology measurements across laboratories supported by sensory trials have revealed that plant proteins increased oral friction due to particle-like protein-protein aggregation and jamming as well as interaction with saliva, in contrast to dairy proteins²⁻⁶. Such lubrication failure or "delubrication"⁴ highlights a major issue if plant proteins are to be used instead of conventional animal proteins, incurring adverse textural modifications and development of these astringent, drying characteristics would reduce consumer acceptability of plant protein-rich foods."

They also highlight a lot of their own papers, but forget to mention other literature in which opposite results are sometimes shown. (for example: whey protein microgels have also been shown to give high friction, whey protein particles do not always show shear thinning behavior. At high concentrations, they have also been shown to give shear thickening behavior..).

****Response.** We see the point of the reviewer. We have included further references where microgels have shown varying degree of lubricity. **Lines 68-72** in the revised manuscript now read as "Animal protein-based, polysaccharide as well as synthetic microgels such as whey protein-based, polyacrylic acid-based and carrageenan-based microgels have demonstrated varying degree of lubricity¹⁰⁻¹² depending upon volume fraction and elasticity of particles entrained in the contact. Often such lubrication mechanism has been governed by viscosity modification with microgels acting as physical surface separators^{11,12} or by so-called (often debated) ball bearing mechanisms^{13,14}."

Figure 4 seems to be the most important results, but the symbols are so large that it is difficult to separate the results. The results are also not discussed and explained well. The results of the emulsion seems rather weird. Even though they mention that this has been shown before, this is not a normal behavior. Even with coalescence, emulsions do not show this behavior.

****Response.** We agree with the reviewer. We have now reduced symbol size and split up the volume fractions to create 12 individual graphs in **revised Fig. 5** (previously Fig. 4 in the original manuscript) that allows for much better clarity on the observed differences between native protein and microgels and also highlight the enhanced lubrication of the microgels. We have analysed the speeds at which the emulsion was entrained for more accurate and clearer

comparisons as the curve was likely a limitation of the machine itself at lower entrainment speeds.

Fig 5] Stribeck curves in hard-soft contact surfaces in presence of plant protein microgels. Tribological performance of steel ball on PDMS surfaces in the presence of plant protein microgels, native plant protein (matched protein content for $\Phi = 70$ vol% with numbers displayed relating to total protein content) or oil-in-water emulsion. Friction coefficient (μ) as a function of entrainment speed (U) scaled with high rate viscosity ($\eta_{\infty} = 1000 \text{ s}^{-1}$) in the presence of plant protein microgels prepared using (a1-3) pea protein concentrate to form a 15.0 wt% total protein microgel, (PPM15), (b1-3) potato protein isolate to form a 5.0 wt% total protein microgel (PoPM5), (c1-3) potato protein isolate to form a 10.0 wt% total protein microgel, (PoPM10), and (d1-3) using a mixture of pea protein concentrate at 7.5 wt% total protein and potato protein isolate at 5.0 wt% total protein microgel (PPM7.5:PoPM5) with 1, 2 and 3 showing increased volume fractions from 10 to 70 vol%, respectively. Frictional responses of the plant proteins at the highest concentration and 20 wt% oil-in-water emulsion (O/W emulsion) and buffer are included in each graph (a-d) as controls. Results are plotted as average of six repeat measurements on triplicate samples ($n = 6 \times 3$) with error bars representing standard deviations. Statistical comparison of mean at 0.1 Pa m is shown in **Supplementary Table S3**. Original friction coefficient versus entrainment speed curves for the microgel dispersions at each volume fractions are shown in **Supplementary Fig. S7**.

Although the mathematical model is a nice addition, they do not really couple equation 11 to the results? What information do they get from it? And why only show this for two of the samples?

****Response.** This is worth pointing that Equation 11 has been related to the results in Fig 6 (previously Fig. 5 in the original submission) where we plot the fitted theoretical model (eq. 11) against experimental data and we supply a discussion. The model interpretation relates to the specific height of the proteins under shear and propose that the model provides an intuitive explanation for the friction coefficients measured. We have now also included the mixed protein microgel in the **revised Figure 6a** again showing that irrespective of the protein type microgels show similar lubricity to that of the emulsions unlike the native proteins with high frictional dissipation.

Fig 6] Mechanism of lubrication performance of plant protein microgels in hard-soft contact surfaces. Tribological performance of steel ball on PDMS contact surfaces showing (a) theoretical modelling of lubrication performance at a load of 2N of exemplar plant protein microgels (pea, potato and mixed pea and potato microgel) showing close resemblance to the emulsions as opposed to the large friction coefficients obtained in presence of the native protein. Here the dashed lines show the best theoretical fit using **equation 11** and (b) load dependency of microgels as compared to the native protein (matched protein content for $\Phi = 70$ vol%) with 20:80 O/W emulsion as control with (c) schematic illustration of microgel performance as compared to native protein in hard-soft contacts. Friction coefficient (μ) is plotted as a function of entrainment speed (U). Results are plotted as average of three repeat measurements on triplicate measurements ($n = 3 \times 3$) with error bars representing standard deviations.

Overall, the manuscripts contains a high level of speculation of the discussions without a systematic experimental approach to proof some of these aspects. Too many different hypothesis have been introduced without clear results. Therefore, I believe the level is not sufficient enough for a high impact journal as Nature communications.

****Response.** We have now addressed all the specific and valuable concerns raised by the reviewer in the revised manuscript with additional experiments on size, shape, elasticity of the microgels well as mathematical modelling on the mixed protein sample. Therefore, we believe this novel, timely work supported by robust experimental and theoretical analysis meets the high quality standards of *Nature Communications*, which also resonates with the supportive views of the other two reviewers who recommend publication.

Reviewer #3 (Remarks to the Author):

Without a doubt the authors have a strong expertise in the field of food colloids, rheological and tribological characterisation of foods and numerical methods. As a consequence I highly appreciate the experimental approach in the present study. The theoretical considerations on lubrication performance are a significant contribution to the field, which has not been published before. However, the innovative character of plant protein microgel utilisation should be elaborated in more detail. Tribological characterisation of plant proteins as well as microgel characterisation of dairy based microgels has already been published by the authors. Plant protein gelation, especially for pea protein, is also well described in several publications. So the reason, why findings of the present study are not on hand from these investigations and the knowledge gap should be elaborated in more detail. I agree that literature on plant protein microgel dispersions is scarce, but authors should explain the scientific innovation justifying publication in Nature Communications in more detail.

****Response.** We thank the reviewer for appreciating the experimental work and the theoretical modelling highlighting the original contribution. The innovative contribution is that we demonstrate that by converting delubricating and astringent plant proteins into microgels, lubrication performance can be enhanced similar to those of emulsions irrespective of the plant protein type, which is evidenced for the first time using a combination of complementary experimental and theoretical approaches.

We present a number of innovations and report fundamental findings, which have not been evidenced in literature to date:

- 1) How plant-based microgels lubricate effectively combining bulk, surface and microstructural approaches and the understanding of the mechanisms of reduction in friction observed using numerical modelling has not been shown before in literature.
- 2) We have now also conducted additional analysis on stability as well as sedimentation to provide more information about their resistance and adaptability to environment (**Supplementary Figure S1, Supplementary Table S1, Supplementary Table S2**) which make these microgels more functional compared to aggregated non-microgelled plant proteins, highlighting the importance of using microgelation as a platform to improve functionality of plant proteins. We thus present a feasible method to standardise plant proteins which behave similarly tribologically (Fig. 6) with such unprecedented ultra-low friction when microgelled and thus can be applied to range of plant proteins. Each are considered to be different in properties (Line 78-85) one is highly soluble, the other is of poor solubility (Lines 92-98), high or low protein concentrations were also used which are important in lubrication/friction but we show that microgels irrespective of concentration or solubility are highly effective in improving lubrication functionality to offer next generation plant-based ingredients.
- 3) Performance of an O/W emulsion to show similarities in friction to plant protein microgels has never been reported in literature.

We have clarified this now in the revised manuscript.

Lines 47-56 in the revised manuscript now read as “It is now well-evidenced that one of the primary barriers to adoption of plant protein is their negative “astringent” *i.e.* dry, puckering, non-juicy perception. Oral lubrication has shown to be crucial to understand, and serves as a

well-acknowledged *in vitro* proxy for quantifying friction-related mouthfeel characteristics¹. Multiscale tribology measurements across laboratories supported by sensory trials have revealed that plant proteins increased oral friction due to particle-like protein-protein aggregation and jamming as well as interaction with saliva, in contrast to dairy proteins²⁻⁶. Such lubrication failure or “delubrication”⁴ highlights a major issue if plant proteins are to be used instead of conventional animal proteins, incurring adverse textural modifications and development of these astringent, drying characteristics would reduce consumer acceptability of plant protein-rich foods.”

Lines 68-72 which read as “Animal protein-based, polysaccharide as well as synthetic microgels such as whey protein-based, polyacrylic acid-based and carrageenan-based microgels have demonstrated varying degree of lubricity¹⁰⁻¹² depending upon volume fraction and elasticity of particles entrained in the contact. Often such lubrication mechanism has been governed by viscosity modification with microgels acting as physical surface separators^{11,12} or by so-called (often debated) ball bearing mechanisms^{13,14}.”

Lines 77-84 in the revised manuscript now read as “Additionally, plant proteins also face intra-variability within the protein type, where both natural climate conditions and industrial processing can result in non-standardised proteins with a range of poor solubility, limited functionality and poor hydrating ability^{7,8} resulting in oral dryness. These present major challenges for plant proteins to be used as a food ingredient where microgelation can be a novel structuring platform to standardize and overcome these oral frictional hurdles, which is the key question this study answers. Therefore, lubrication evidences on microgelation of plant proteins are imperative before such microgels can be applied as a highly functional, pleasurable ingredient for designing next generated plant protein-reformulated foods⁹.”

Lines 85-87, which read as “Herein, we demonstrate using a combination of experimental and theoretical approaches that engineering physically cross-linked, soft, sub-micron sized plant-based microgels offers superior lubrication performance as compared to their parent native protein counterparts for the first time.”

Apart from some minor issues listed below I have some more fundamental points, which should be addressed:

Motivation for the setup is not entirely clear to me. Why did the authors decide for a blend of the two proteins and why did the authors vary the protein content, if they want to compare the plant proteins.

****Response.** The reviewer has raised an important question. A blend of the proteins was investigated to determine any synergistic or detrimental effect on lubricity. The latter approach also provides a way to enhance the amino acid profile of possible formulations through protein complementation, a crucial consideration in current formulation design of non-animal protein diets. WE have now included some statements in this direction.

Lines 89-97 in the revised manuscript read as “This study utilised pea protein and potato protein as typical exemplar plant proteins which take the form of globular multimer storage proteins. Pea proteins are composed mostly of 11S and 7S globulins¹⁵ and frequently reported for being of limited aqueous solubility^{3,8,15} result in high friction and sensorial astringency^{4,16}.

On the other hand, potato proteins are composed of rather unique, globular, glycoprotein *i.e.* patatins¹⁷, with comparatively higher fractions of soluble protein^{3,17,18} but still suffer from astringency issues^{3,16} due to high surface hydrophobicity¹⁹. A blend of the proteins were also investigated to determine any synergistic or detrimental effect on lubricity. The latter approach also provides a way to enhance the amino acid profile of possible formulations through protein complementation, a crucial consideration in non-animal protein diets²⁰.”

If proteins have a quite large difference in denaturation temperature and they do not complex in one particle (line 119), what is the expected benefit of the binary blend and the oscillatory shear rheology experiments of the blend? In line 189 you state that the microgels are sub-micrometric units of the parent hydrogels. This is somehow contradictory to the statement previously cited.

****Response.** We thank the reviewer for this comment. This mixed plant protein system indeed become microgels as can be evidenced clearly in newly added AFM images in **Figure 3d** (see below) in the revised manuscript but have not complexed together and form differently-sized population of microgels. Our question was whether such mixed system detriment the lubrication behaviour. However, as can be seen in the data showing both experimental and numerical fits in **Figure 6a**, even the mixed microgel system (newly added data) show similar lubricity to that of oil-in-water emulsions as a single protein system despite having different elasticity and size range.

Fig 3| Images of plant protein microgels on silicon under buffer. Topographic images and respective histograms showing diameters of aqueous dispersions of protein microgels prepared using (a) pea protein concentrate to form a 15.0 wt% total protein microgel, (PPM15), potato protein isolate to form a 5.0 wt% total protein microgel (PoPM5), potato protein isolate to form a 10.0 wt% total protein microgel, (PoPM10), and using a mixture of pea protein concentrate at 7.5 wt% total protein and potato protein isolate at 5.0 wt% total protein microgel (PPM7.5:PoPM5) obtained by atomic force microscopy (AFM). Most of the microgels are distributed homogeneously on the surface.

Fig 6| Mechanism of lubrication performance of plant protein microgels in hard-soft contact surfaces. Tribological performance of steel ball on PDMS contact surfaces showing (a) theoretical modelling of lubrication performance at a load of 2N of exemplar plant protein microgels (pea, potato and mixed pea and potato microgel) showing close resemblance to the emulsions as opposed to the large friction coefficients obtained in presence of the native protein. Here the dashed lines show the best theoretical fit using **equation 11** and (b) load dependency of microgels as compared to the native protein (matched protein content for $\Phi = 70$ vol%) with 20:80 O/W emulsion as control with (c) schematic illustration of microgel performance as compared to native protein in hard-soft contacts. Friction coefficient (μ) is plotted as a function of entrainment speed (U). Results are plotted as average of three repeat measurements on triplicate measurements ($n = 3 \times 3$) with error bars representing standard deviations.

Furthermore, the authors should explain in more detail, why they chose to compare the microgel particles to native proteins.

****Response.** We thank the reviewer for this comment. This is the primary hypothesis that by converting plant proteins which are inherently delubricating, into microgels, we can make them high performance lubricants. Hence it was essential to compare them against native proteins.

With a high percentage of aggregated and insoluble protein, conclusions like in line 155 ("rheological behaviour of microgels is in sharp contrast to native proteins") appear obvious and the shear-thinning behaviour of plant proteins has been reported before. Also the difference lubrication as discussed in line 230ff. is an obvious consequence of the gelation.

****Response.** We agree with the reviewer and thus have removed this statement.

Line 47, Typo: "it" instead of "it"

Line 49, Typo: "dairy" instead of "diary"

****Response.** We thank the reviewer for highlighting these typos, we have now corrected them and a few more throughout the manuscript.

Line 52: I do not agree, that lubrication issues in plant-based foods are addressed through ultra-processing and result in unsustainable, unhealthy and undesirable foods. I am convinced that the authors would also not agree to such a generic statement and recommend to either give specific examples or re-phrase. Personally I would completely remove this aspect, since the materials science aspect of the poor lubrication is sufficient to justify the research.

****Response.** We agree with the reviewer that this statement was generic in the original submission and have removed this from the text.

Line 96: I recommend to add an explanation, why the percentage of protein has been varied when using different proteins.

****Response.** The percentages of protein was varied to alter the elasticity of the parent gels and subsequently the microgels as it is known that elasticity might affect the lubricity of microgels. This is now added to the text.

Lines 110-118 in the revised manuscript now read as “Four types of microgels were fabricated utilising a top-down methodology involving thermal gelation-induced physical crosslinking of the *otherwise sedimenting* (Supplementary Fig. S1) plant proteins resulting in a percolating, viscoelastic protein-based hydrogels followed by controlled shearing into microgels (Fig. 1, see detailed preparation in method section). In order to attain different elasticities of these microgels, which are known to be important in lubrication¹², and the high protein concentrations typically found in tribology studies involving high levels of friction³, the concentration of the solutions was adjusted to promote gel formation with varying properties which offer an opportunity to study the microgelation process and its effectiveness for lubrication in plant proteins (see details in method section).”

Line 115, Figure title: one may misunderstand it in a way that the protein content of the ppc has been 15%

****Response.** We thank the reviewer for pointing this out, the text has now been changed in all figure captions to improve clarity that the microgels were indeed made using protein powders with protein of 5,10,15 or 7.5 and 15 wt% protein.

Line 116: Add "particles" after "Individual microgels"

****Response.** We have now used “microgels” throughout the manuscript but where relevant in general terms have named them as particles.

Line 125: What is the reason for the variation in DLS. Later on in the discussion authors should comment on the impact of the size on DLS analysis.

****Response.** The variation in size is dependant on native protein size, solubility and water holding capacity.

This is now highlighted in **Lines 157-159** in the revised manuscript, which read as “Such d_H values are typical to other reported microgels from both animal and plant protein-based sources^{12,30-32} with their differences in size dependant on protein size, solubility and water holding capacity.”

The importance of size has been mentioned in and included in the “Theoretical mechanism of lubrication (**Lines 449-499**)” (**Table 1**). The size was also used to explain such behaviour in biomimetic tongue-like surface (**Lines 476-486**). In general microgels range in size from 70 to 200 nm and all of them provide ultralow friction despite differences in sizes.

Is the motion of the particles analysed in DLS exclusively based on electrophoretic motion?

****Response.** Yes, the motion was analysed based on electrophoretic mobility. However, the size measured using DLS was similar to those evidenced by AFM. This is further detailed in Lines 168-198, which read as **Lines 168-198** in the revised manuscript now read as “To investigate microgel morphology in more detail, atomic force microscopy (AFM) was employed to image fully hydrated microgels, shown in Fig 3 with corresponding particle size distributions. All microgels are in the same general size range of 50-200 nm as found by DLS. (Fig. 2b).

The mixed PPM:PoPM (Fig 3d) system displays two distinctly sized population of microgels which could be the result of sub-unit protein complexes forming from pea and potato protein as discussed previously in DLS. PoPM5 (Fig 3b) is remarkably similar in size in both techniques, 67 nm (DLS) vs 73 nm (AFM). In the other samples the AFM size was slightly smaller than DLS; for PoPM10 (Fig 3c) the diameters DLS:AFM were 132 nm:109 nm, and for the mixed PPM:PoPM were 90 nm: 78 nm for the first peak, and 260 nm: 158 nm for the second. This reduction in size can be explained by noting that the loosely structured brush-like features extends from the surface of microgels and occupies a larger hydrodynamic volume, we speculate the increased overestimation from DLS could be related to the hydration shell whilst force microscopy measures the protein core, this explains the similar size of PoPM5 and larger size of PoPM10 as more protein have an extended influence on hydrodynamic diameter. PPM15 (Fig 3a) showed a larger discrepancy, 204 nm (DLS) vs 79 nm (AFM). This could be explained by the presence of a small number of polydisperse large particles or aggregates in this sample which could skew the DLS measurement to a higher value (these aggregates were not seen in the distributions of PoPM5 and PoPM10) or show the presence of a highly hydrated shell spanning far from the core. Shape analysis (Fig S2) showed the majority of particles were spherical or near spherical with an aspect ratio < 2:1, although with increasing aspect ratio with size, which is probably explained by the larger particles being aggregates. Overall microgels take on a smooth and spherical shape showing convex spreading on the surface (Fig 3a-d) which has also been observed for synthetic and whey protein microgels in previous studies^{12,29}.”

Line 144: Please explain what you mean by spreading-like behaviour and which result you are exactly referring to.

****Response.** We are referring here to AFM results, This is now clarified in **Lines 196-198** which read as “Overall microgels take on a smooth and spherical shape showing convex spreading on the surface (Fig 3a-d) which has also been observed for synthetic and whey protein microgels in previous studies^{12,29}.”

Line 195: What does "stark" mean? I am sorry, but I have never read this word before in a manuscript.

****Response.** We have removed this from the text.

Line 218. I assume that drop size in the emulsion plays a major role in tribology. How was it adjusted to microgel size. At least I recommend to report the drop size distribution.

****Response.** We agree and have added the droplet size distribution of the emulsion as Supplementary Figure 10. Indeed size can have an influence but the mechanism of emulsion tribology is often linked to droplet coalescence induced oil films rather than intact emulsion droplets. As highlighted in **Table 1** on calculation and in **Lines 447—480**, both emulsions and microgel will deform under tribological shear, get entrained and reduce friction, through by different mechanisms as described in the text.

Figure 4: too crowded for me, it is not possible to properly compare the different samples. Furthermore I would recommend to indicate the different lubrication regimes. In the presentation comparison is done at a specific volume fraction. MAYbe the whole figure should be adapted in that way, so that the figure matches the story line in the text.

****Response.** We agree with the reviewer. We have now reduced symbol size and split up the volume fractions to create 12 individual graphs in **revised Fig. 5** (previously Fig. 4 in the original manuscript) that allows for much better clarity on the observed differences between native protein and microgels and also highlight the enhanced lubrication of the microgels. We have analysed the speeds at which the emulsion was entrained for more accurate and clearer comparisons as the curve was likely a limitation of the machine itself at lower entrainment speeds.

Fig 5] Stribeck curves in hard-soft contact surfaces in presence of plant protein microgels. Tribological performance of steel ball on PDMS surfaces in the presence of plant protein microgels, native plant protein (matched protein content for $\Phi = 70$ vol% with numbers displayed relating to total protein content) or oil-in-water emulsion. Friction coefficient (μ) as a function of entrainment speed (U) scaled with high rate viscosity ($\eta_{\infty} = 1000 \text{ s}^{-1}$) in the presence of plant protein microgels prepared using (**a1-3**) pea protein concentrate to form a 15.0 wt% total protein microgel, (PPM15), (**b1-3**) potato protein isolate to form a 5.0 wt% total protein microgel (PoPM5), (**c1-3**), potato protein isolate to form a 10.0 wt% total protein microgel, (PoPM10), and (**d1-3**) using a mixture of pea protein concentrate at 7.5 wt% total protein and potato protein isolate at 5.0 wt% total protein microgel (PPM7.5:PoPM5) with 1, 2 and 3 showing increased volume fractions from 10 to 70 vol%, respectively. Frictional responses of the plant proteins at the highest concentration and 20 wt% oil-in-water emulsion (O/W emulsion) and buffer are included in each graph (**a-d**) as controls. Results are plotted as average of six repeat measurements on triplicate samples ($n = 6 \times 3$) with error bars representing standard deviations. Statistical comparison of mean at 0.1 Pa m is shown in **Supplementary Table S3**. Original friction coefficient versus entrainment speed curves for the microgel dispersions at each volume fractions are shown in **Supplementary Fig. S7**.

Line 453: "resulted increases" - structure of the sentence not clear

****Response.** We thank the reviewer for pointing this out and rectified this mistake in the revised manuscript.

Line 455: please discuss in more detail the nature of the aggregates

****Response.** We believe aggregates is not the correct terminology, we have rectified this and included more explanation on this. Nevertheless, the particle size and shape have been detailed in the sizing second with DLS and AFM as mentioned above.

Lines 503-515 in the revised manuscript read as “This technique does present several differences arising from the properties of the tongue-surface specifically, its wettability, topography and deformability which may incur high frictional sensitivity in the boundary friction. In particular PoPM5 $\Phi = 10$, PoPM10 $\Phi = 10$ and PPM7.5:PoPM5 $\Phi = 70$ were found to have frictions higher than O/W at lower V_R ($p > 0.05$). We speculate a higher δR^* and lower d_H may contribute to an initial ineffective lubrication for potato protein microgels as there may not be sufficient levels of microgel for successful entrainment. However, at higher V_R the friction reduces where a non-significant difference in μ to those in O/W emulsion is observed ($p > 0.05$). For PPM7.5:PoPM5.0, an increase in volume fraction results in increased friction. As this dispersion is made up of two sets of microgels of different moduli and size, a build-up may occur at higher concentrations at the papillae promoting a jamming behaviour opposed to uniformly hydrated ball-bearing-type lubrication if microgel of one protein type³³ is present. This phenomenon is not observed in glass and PDMS as is likely the flat surface contact allow separate microgels to flow over one another without disruption improving lubricity.”

Line 513: With an oil load of 20% the emulsion is not a high fat emulsion in my opinion.

****Response.** We agree with this and have removed the term “high fat” and replaced with 20:80 O/W emulsion

Line 582ff. : Why did the authors use two different homogenisers? This will inevitably lead to differences in microgel particle structure as well as rheological and tribological behaviour.

****Response.** This was a mistake, only the PANDA two-stage homogeniser was used for producing the final microgels, this is corrected in the manuscript.

Line 586: Please decide, if you name the final microgel, particle or microgel particle and consequently use the term throughout the text.

****Response.** We have changed this in text and have standardised this throughout the manuscript calling it “microgel”.

Line 600/631/675: It is not clear whether multiple preparations of the microgel systems have been performed. Means and standard deviations should only be provided, if they are based on true replicates rather than analytical replicates.

****Response.** This information has now been added in the methods section, all samples were of multiple batches repeated on separate days. Also the captions are corrected in the revised manuscript.

Lines 782-783 in the revised manuscript read as “All results are reported as means and standard deviations on at least three measurements carried out on three independent samples prepared on separate days.”

Line 625: During gel preparation the time was 30 min, in rheology the authors reduced to 10 min. This may heavily affect gel structure and hinders a comparison.

****Response.** This was a mistake and has been rectified, we have also included the temperature ramp data (Supplementary Figure S3) of the parent gel formation from which the microgels were fabricated by controlled shearing, this indeed replicates the time-temperature process used for fabricating the microgels outside the rheometer.

References

- 1 Sarkar, A. & Krop, E. M. Marrying oral tribology to sensory perception: a systematic review. *Current Opinion in Food Science* **27**, 64-73, doi:<https://doi.org/10.1016/j.cofs.2019.05.007> (2019).
- 2 Zembyla, M. *et al.* Surface adsorption and lubrication properties of plant and dairy proteins: A comparative study. *Food Hydrocolloids* **111**, 106364, doi:<https://doi.org/10.1016/j.foodhyd.2020.106364> (2021).
- 3 Kew, B., Holmes, M., Stieger, M. & Sarkar, A. Oral tribology, adsorption and rheology of alternative food proteins. *Food Hydrocolloids* **116**, 106636, doi:<https://doi.org/10.1016/j.foodhyd.2021.106636> (2021).
- 4 Vlădescu, S.-C. *et al.* Protein-induced delubrication: How plant-based and dairy proteins affect mouthfeel. *Food Hydrocolloids* **134**, 107975, doi:<https://doi.org/10.1016/j.foodhyd.2022.107975> (2023).
- 5 Liamas, E., Connell, S. D. & Sarkar, A. Frictional behaviour of plant proteins in soft contacts: unveiling nanoscale mechanisms. *Nanoscale Advances*, doi:10.1039/D2NA00696K (2023).
- 6 Pang, Z., Tong, F., Jiang, S., Chen, C. & Liu, X. Particle characteristics and triborheological properties of soy protein isolate (SPI) dispersions: Effect of heating and incorporation of flaxseed gum. *International Journal of Biological Macromolecules* **232**, 123455, doi:<https://doi.org/10.1016/j.ijbiomac.2023.123455> (2023).
- 7 Ma, K. K. *et al.* Functional Performance of Plant Proteins. **11**, 594 (2022).
- 8 Lam, A. C. Y., Can Karaca, A., Tyler, R. T. & Nickerson, M. T. Pea protein isolates: Structure, extraction, and functionality. *Food Reviews International* **34**, 126-147, doi:10.1080/87559129.2016.1242135 (2018).
- 9 Sim, S. Y. J., SRV, A., Chiang, J. H. & Henry, C. J. Plant Proteins for Future Foods: A Roadmap. **10**, 1967 (2021).
- 10 de Vicente, J., Stokes, J. R. & Spikes, H. A. Soft lubrication of model hydrocolloids. *Food Hydrocolloids* **20**, 483-491, doi:<https://doi.org/10.1016/j.foodhyd.2005.04.005> (2006).
- 11 Garrec, D. A. & Norton, I. T. Kappa carrageenan fluid gel material properties. Part 2: Tribology. *Food Hydrocolloids* **33**, 160-167, doi:<https://doi.org/10.1016/j.foodhyd.2013.01.019> (2013).
- 12 Andablo-Reyes, E. *et al.* Microgels as viscosity modifiers influence lubrication performance of continuum. *Soft Matter* **15**, 9614-9624, doi:10.1039/C9SM01802F (2019).
- 13 Liu, K., Stieger, M., van der Linden, E. & van de Velde, F. Effect of microparticulated whey protein on sensory properties of liquid and semi-solid model foods. *Food Hydrocolloids* **60**, 186-198, doi:<https://doi.org/10.1016/j.foodhyd.2016.03.036> (2016).
- 14 Liu, K., Tian, Y., Stieger, M., van der Linden, E. & van de Velde, F. Evidence for ball-bearing mechanism of microparticulated whey protein as fat replacer in liquid and semi-solid multi-component model foods. *Food Hydrocolloids* **52**, 403-414, doi:<https://doi.org/10.1016/j.foodhyd.2015.07.016> (2016).
- 15 S. Tomoskozi, R. H. a. O. B. Isolation and study of the functional properties of pea proteins. *Nahrung/Food* **45**, 399-401 (2001).

- 16 Tanger, C. *et al.* Influence of Pea and Potato Protein Microparticles on Texture and Sensory Properties in a Fat-Reduced Model Milk Dessert. *ACS Food Science & Technology* **2**, 169-179, doi:10.1021/acscfoodscitech.1c00394 (2022).
- 17 Waglay, A., Karboune, S. & Alli, I. Potato protein isolates: Recovery and characterization of their properties. *Food Chemistry* **142**, 373-382, doi:<https://doi.org/10.1016/j.foodchem.2013.07.060> (2014).
- 18 Schmidt, J. M. *et al.* Gel properties of potato protein and the isolated fractions of patatins and protease inhibitors – Impact of drying method, protein concentration, pH and ionic strength. *Food Hydrocolloids* **96**, 246-258, doi:<https://doi.org/10.1016/j.foodhyd.2019.05.022> (2019).
- 19 Mao, C., Wu, J., Zhang, X., Ma, F. & Cheng, Y. Improving the Solubility and Digestibility of Potato Protein with an Online Ultrasound-Assisted PH Shifting Treatment at Medium Temperature. **9**, 1908 (2020).
- 20 Dimina, L., Rémond, D., Huneau, J.-F. & Mariotti, F. Combining Plant Proteins to Achieve Amino Acid Profiles Adapted to Various Nutritional Objectives—An Exploratory Analysis Using Linear Programming. **8**, doi:10.3389/fnut.2021.809685 (2022).
- 21 Lie-Piang, A., Braconi, N., Boom, R. M. & van der Padt, A. Less refined ingredients have lower environmental impact – A life cycle assessment of protein-rich ingredients from oil- and starch-bearing crops. *Journal of Cleaner Production* **292**, 126046, doi:<https://doi.org/10.1016/j.jclepro.2021.126046> (2021).
- 22 Pampuri, A. *et al.* Environmental Impact of Food Preparations Enriched with Phenolic Extracts from Olive Oil Mill Waste. **10**, 980 (2021).
- 23 Ashokkumar, K., Tar'an, B., Diapari, M., Arganosa, G. & Warkentin, T. D. Effect of Cultivar and Environment on Carotenoid Profile of Pea and Chickpea. **54**, 2225-2235, doi:<https://doi.org/10.2135/cropsci2013.12.0827> (2014).
- 24 Heller, M. C. & Keoleian, G. A. Beyond meat's beyond burger life cycle assessment: a detailed comparison between. (2018).
- 25 Saerens, W., Smetana, S., Van Campenhout, L., Lammers, V. & Heinz, V. Life cycle assessment of burger patties produced with extruded meat substitutes. *Journal of Cleaner Production* **306**, 127177, doi:<https://doi.org/10.1016/j.jclepro.2021.127177> (2021).
- 26 Aganovic, K. *et al.* Pilot scale thermal and alternative pasteurization of tomato and watermelon juice: An energy comparison and life cycle assessment. *Journal of Cleaner Production* **141**, 514-525, doi:<https://doi.org/10.1016/j.jclepro.2016.09.015> (2017).
- 27 Baune, M.-C. *et al.* Meat hybrids—An assessment of sensorial aspects, consumer acceptance, and nutritional properties. **10**, doi:10.3389/fnut.2023.1101479 (2023).
- 28 Liamas, E., Connell, S. D., Zembyla, M., Ettelaie, R. & Sarkar, A. Friction between soft contacts at nanoscale on uncoated and protein-coated surfaces. *Nanoscale* **13**, 2350-2367, doi:10.1039/D0NR06527G (2021).
- 29 Aufderhorst-Roberts, A. *et al.* Nanoscale mechanics of microgel particles. *Nanoscale* **10**, 16050-16061, doi:10.1039/C8NR02911C (2018).
- 30 Araiza-Calahorra, A. & Sarkar, A. Pickering emulsion stabilized by protein nanogel particles for delivery of curcumin: Effects of pH and ionic strength on curcumin retention. *Food Structure* **21**, 100113, doi:<https://doi.org/10.1016/j.foostr.2019.100113> (2019).

- 31 Zhang, S., Holmes, M., Ettelaie, R. & Sarkar, A. Pea protein microgel particles as Pickering stabilisers of oil-in-water emulsions: Responsiveness to pH and ionic strength. *Food Hydrocolloids* **102**, 105583, doi:<https://doi.org/10.1016/j.foodhyd.2019.105583> (2020).
- 32 Aery, S. *et al.* Ultra-stable liquid crystal droplets coated by sustainable plant-based materials for optical sensing of chemical and biological analytes. *Journal of Materials Chemistry C*, doi:10.1039/D3TC00598D (2023).
- 33 Wan, J., Ningtyas, D. W., Bhandari, B., Liu, C. & Prakash, S. Oral perception of the textural and flavor characteristics of soy-cow blended emulsions. *Journal of Texture Studies* **53**, 108-121, doi:<https://doi.org/10.1111/jtxs.12641> (2022).

Reviewers' Comments:

Reviewer #1:

Remarks to the Author:

The authors have addressed all my comments in detail, so I agree to publish this paper as it is.

Reviewer #4:

Remarks to the Author:

The manuscript titled "Transforming sustainable plant proteins into high performance lubricating microgels" explore the lubrication performance of various microgel systems based on various plant-based proteins. The manuscript provides an in-depth characterization of the microgel particles structure and characteristics while the lubrication performance was analyzed experimentally as well as theoretically using modeling. The research presents a very high level of scientific understanding combining experimental and modeling results.

The authors have done a significant improvement to the original manuscript and answered most of the reviewer's comments. However, the reviewer feels that the innovation and significance of this research were not emphasized enough justifying publication in Nature Communications.

Moreover, to emphasize the reliability and applicability of the results the authors should address the relation between the mimetic tongue system and real sensory analysis?

REVIEWERS' COMMENTS

Reviewer #1 (Remarks to the Author):

The authors have addressed all my comments in detail, so I agree to publish this paper as it is.

****Response:** We thank the review for recommending acceptance.

Reviewer #4 (Remarks to the Author):

The manuscript titled “Transforming sustainable plant proteins into high performance lubricating microgels” explore the lubrication performance of various microgel systems based on various plant-based proteins. The manuscript provides an in-depth characterization of the microgel particles structure and characteristics while the lubrication performance was analyzed experimentally as well as theoretically using modeling. The research presents a very high level of scientific understanding combining experimental and modeling results. The authors have done a significant improvement to the original manuscript and answered most of the reviewer’s comments. However, the reviewer feels that the innovation and significance of this research were not emphasized enough justifying publication in Nature Communications.

****Response:** We thank the reviewer for echoing the very high level of scientific understanding this manuscript provides combining modelling and experimental results using in-depth characterization using multiple complementary techniques. We have already clearly detailed the innovation and significance of these results throughout the submitted revised manuscript in Introduction and Discussion that indeed this is the first study on microgelation of plant proteins showing significant lubricity versus a native plant protein and resemblance to fat emulsions without using any fat. This has a huge significance to offer a new platform of plant protein-based food ingredients to enhance palatability and functionality, thus, the improved design of next generation protein-based, healthy, tasty and sustainable diets in order to accelerate the transition to plant-based foods.

Moreover, to emphasize the reliability and applicability of the results the authors should address the relation between the mimetic tongue system and real sensory analysis?

****Response.** We see the point of the reviewer. We had already improved the introduction to give clarity on why *in vitro* tribology was used in the previously revised submitted version. Nevertheless, we have further made this explicit and also highlighted that sensory evaluation is a key necessary future step. Following text in the revised manuscript state:

Line 488-495: Oral tribology has provided significant advances in friction mediated sensory responses, which is supported by over a decade of correlating tribology to real sensory attributes^{6,43,44}. In oral tribology, the paradigm has historically been using smooth, hydrophobic, PDMS tribopairs of high elastic modulus (~ 2 MPa) as a surface to represent human tongue, however such materials differ in their wettability, contact pressure and topography from a real human tongue^{45,46}, thus hindering true friction-sensory correlation in complex soft materials. In the pursuit of improving accuracy and reliability of *in vitro* oral

mouthfeel measurements, specific attention is given on development of biorelevant surfaces⁴⁵.

Line 616-618: Nonetheless, understanding the real friction-mediated sensory analysis of plant protein microgels remains as a necessary undertaking, which is outside the scope of this study.

Line 639-642: Ultimately, converting native plant proteins into microgels offers a facile platform to solve friction-related issues and combining this mechanistic work with sensory studies in the future will allow rapid transition from animal to palatable plant protein-based diets to promote planetary health.